# High-performance and stable photoelectrochemical water splitting cell with organic-photoactive-layer-based photoanode

Je Min Yu[1,6], Jungho Lee[1,2,6], Yoon Seo Kim [3,6], Jaejung Song[3], Jiyeon Oh[1], Sang Myeon Lee[1], Mingyu Jeong[1], Yongseon Kim[1], Ja Hun Kwak[1], Seungho Cho[3 ✉], Changduk Yang [1,4 ✉] & Ji-Wook Jang [1,5 ✉]

Considering their superior charge-transfer characteristics, easy tenability of energy levels, and low production cost, organic semiconductors are ideal for photoelectrochemical (PEC) hydrogen production. However, organic-semiconductor-based photoelectrodes have not been extensively explored for PEC water-splitting because of their low stability in water. Herein, we report high-performance and stable organic-semiconductors photoanodes consisting of p-type polymers and n-type non-fullerene materials, which is passivated using nickel foils, GaIn eutectic, and layered double hydroxides as model materials. We achieve a photocurrent density of 15.1 mA cm$^{-2}$ at 1.23 V vs. reversible hydrogen electrode (RHE) with an onset potential of 0.55 V vs. RHE and a record high half-cell solar-to-hydrogen conversion efficiency of 4.33% under AM 1.5 G solar simulated light. After conducting the stability test at 1.3 V vs. RHE for 10 h, 90% of the initial photocurrent density are retained, whereas the photoactive layer without passivation lost its activity within a few minutes.

[1] Department of Energy Engineering, School of Energy and Chemical Engineering, Ulsan National Institute of Science and Technology (UNIST), Ulsan 44919, Republic of Korea. [2] Department of Chemistry, Purdue University, West Lafayette, IN 47907, USA. [3] Department of Materials Science and Engineering, Ulsan National Institute of Science and Technology (UNIST), Ulsan 44919, Republic of Korea. [4] School of Energy and Chemical Engineering, Perovtronics Research Center, Low Dimensional Carbon Materials Center, Ulsan National Institute of Science and Technology (UNIST), Ulsan 44919, Republic of Korea. [5] Emergent Hydrogen Technology R&D Center, Ulsan National Institute of Science and Technology (UNIST), Ulsan 44919, Republic of Korea. [6]These authors contributed equally: Je Min Yu, Jungho Lee, Yoon Seo Kim. ✉email: scho@unist.ac.kr; yang@unist.ac.kr; jiwjang@unist.ac.kr

The photoelectrochemical (PEC) water splitting technology is considered one of the most promising $H_2$ production methods because it utilizes the unlimited energy source of solar light and does not emit $CO_2$[1]. In addition to their sufficient earth abundance and stability in water, photoelectrode materials must exhibit solar-to-hydrogen (STH) conversion efficiency of above 10% for their commercial viability[2]. Since the first successful demonstration of water splitting with a $TiO_2$ photoanode by Honda and Fujishima[3], extensive research has been focused on strategies to maximize the STH efficiency of inorganic photocatalysts, such as $SrTiO_3$, $NaTaO_3$, $WO_3$, $Fe_2O_3$, $BiVO_4$, and $Ta_3N_5$ because they are typically earth-abundant compounds and are stable in water[4–10].

To obtain a high STH conversion efficiency, the conduction and valence band positions of a photocatalyst should be more negative than the hydrogen evolution potential and more positive than the oxygen evolution potential, respectively, with an optimum bandgap of 1.6–1.8 eV[11]. However, for more than 35 years, no inorganic material has been found to meet all these requirements. Furthermore, due to their intrinsically poor charge-transfer characteristics, the half-cell STH conversion efficiencies ($\eta_{\text{half-STH}}$, the STH efficiency considering the applied external bias) of inorganic semiconductors are still in the range of 0.01–2.5% under AM 1.5 G solar simulated light[12–16].

Organic semiconductors comprise carbon, which is one of the most earth-abundant elements, and exhibit charge-transfer characteristics superior to their inorganic counterparts. In particular, band positions and bandgaps of organic semiconductors are readily tuned, and thus, the theoretical maximum STH efficiency of PEC water splitting is over 30%[17,18]. Moreover, an organic PEC system can work without applying bias by band position engineering, indicating that a solar cell is not required, which has been considered a requirement for PEC commercialization. In addition, organic semiconductors can be used for both photoanodes and photocathodes by simply converting their order of deposition[19–22]. Despite these merits, organic semiconductors have not attracted attention as photoactive materials for PEC water splitting due to their low stability in aqueous solutions[20]. Several organic semiconductors were applied for photocathodes, since the first demonstration of photocathodic hydrogen production by Shirakawa et al. with polyacetylene in 1981[23–26], and only few organic photoactive-layer-based photoanodes have been reported. For example, Abe et al. showed the possibility that organic semiconductor can be applied for a photoanode with a composite of 3,4,9,10-perylenetetracarboxylic acid bisbenzimidazole (n-type semiconductor) and cobalt(II) phthalocyanine(p-type semiconductor)[27]. Bornoz et al.[28] and Dai et al.[29] showed that the poly[benzimidazobenzophenanthroline] and fluorine-dibenzothiophene-S,S-dioxide-based conjugated polymer can be utilized as photoanodes for direct solar water oxidation, respectively. Ruan et al.[30] and Peng et al.[31] showed the possibility of usage of carbon nitride materials as photoanodes. However, these organic photoactive-layer-based photoanodes not only exhibited photocurrent densities of only few microampere scales at 1.23 V vs. reversible hydrogen electrode (RHE) (~100 µA cm$^{-2}$) but half-cell STH conversion efficiencies ($\eta_{\text{half-STH}}$) were also lower than 0.03% so far (Supplementary Table 1 and Supplementary Fig. 1). Furthermore, they lost their performances in a few minutes and even for the case of recent stable organic photoelectrodes stabilized by $TiO_2$ layers, less than 50% initial photocurrent is maintained after 1 h[21].

Herein, we use Ni metal foils to encapsulate an organic photoactive layer, for preventing water intrusion, and GaIn eutectic as a mediator between the organic photoactive layer and Ni foils, for an efficient charge transport. Finally, we synthesize nickel iron layered double hydroxides (NiFe-LDHs) to maximize the charge-separation efficiency and suppress photocorrosion of the organic photoactive layer by surface-reaching holes (Fig. 1). NiFe-LDHs also serve as a catalyst for oxygen evolution reaction by photo-generated hole (OER). Note that we use an archetype of high-performance polymer/non-fullerene as an organic photoactive layer for a photoanode (bulk-heterojunction blend, BHJ of p-type poly[(2,6-(4,8-bis(5-(2-ethylhexyl)thiophen-2-yl)-benzo[1,2-b:4,5-b′]dithiophene))-alt-(5,5-(1′,3′-di-2-thienyl-5′,7′-bis(2-ethylhexyl)benzo[1′,2′-c:4′,5′-c′]dithiophene-4,8-dione)] (denoted as PBDB-T) and n-type 3,9-bis(2-methylene-(3-(1,1-dicyanomethylene)-indanone))-5,5,11,11-tetrakis(4-hexylphenyl)-dithieno[2,3-d:2′,3′-d′]-s-indaceno[1,2-b:5,6-b′]dithiophene (denoted as ITIC[32]), which has distinguished advantages of easy tuning of absorption and electronic energy levels, high photostability, and cost-effectiveness over conventional fullerene-based ones[33]. From the designed organic PEC cell, we achieve a photocurrent density of 15.1 mA cm$^{-2}$ at 1.23 V vs. RHE with an onset potential of 0.55 V vs. RHE and a $\eta_{\text{half-STH}}$ of 4.33% under AM 1.5 G solar simulated light, which is, to our best knowledge, the highest value among those of all reported photoanodes (Supplementary Table 1 and Supplementary Fig. 1). Finally, the stability of the photoanode is tested under 1 sun illumination at 1.3 V vs. RHE. Approximately, 90% of the initial current density is retained after 10 h, while the organic photoactive layer without passivation loses its activity within a few minutes. We also investigate the reason for decreases in their performance values in terms of chemical stability, photocorrosion induced by the surface charge, and photostability of the organic semiconductor itself.

## Results

**Design of organic PEC cell**. In this study, we use an inverted n–i–p configuration of PBDB-T:ITIC-based organic photovoltaics (OPVs) as a photoanode for water oxidation (Fig. 1 and Supplementary Fig. 2). The most important mission here is how an organic semiconductor-based photoactive layer is effectually shielded from water while maintaining an efficient charge transfer from OPVs to water. Deposition of conductive metals on OPVs by physical vapor deposition methods such as sputtering and evaporation can be conducted to achieve this; voids can exist in the deposited thin metal films, through which water can penetrate[34,35]. When a thick film is deposited for a long duration, from several hours to days, to mitigate water insertion, the junction between the metal and organic material can become relatively weak, resulting in the detachment of the metal layer. To completely protect an organic layer from water intrusion, we apply a conductive Ni foil onto organic materials. However, in this case, the contact between Ni metal foils and OPVs becomes much worse, leading to poorer charge transfer as compared to the case of a directly deposited metal layer. To solve this problem, we use a GaIn eutectic as an electrically conductive and flexible connector between Ni foils and OPVs to facilitate efficient charge transport. To further enhance the water splitting performance, we grow NiFe-LDHs directly on Ni foils, which is reported as one of the best performing earth-abundant oxygen evolution reaction catalysts[36]. The detailed process for the photoelectrode fabrication is schematically illustrated with the relevant photographs and explanations in Supplementary Fig. 3 and the Methods part.

**Preparation of organic-photoactive-layer-based photoanodes**. Following the above device design, we first prepare PBDB-T:ITIC-based OPVs, where an organic photoactive layer is spin-casted on the ZnO electron transport layer on top of a UV-ozone-treated indium–tin–oxide (ITO) substrate. Atomic force microscope (AFM) images of the photoactive active layer show that PBDB-T and ITIC are homogenously well-deposited on the ITO

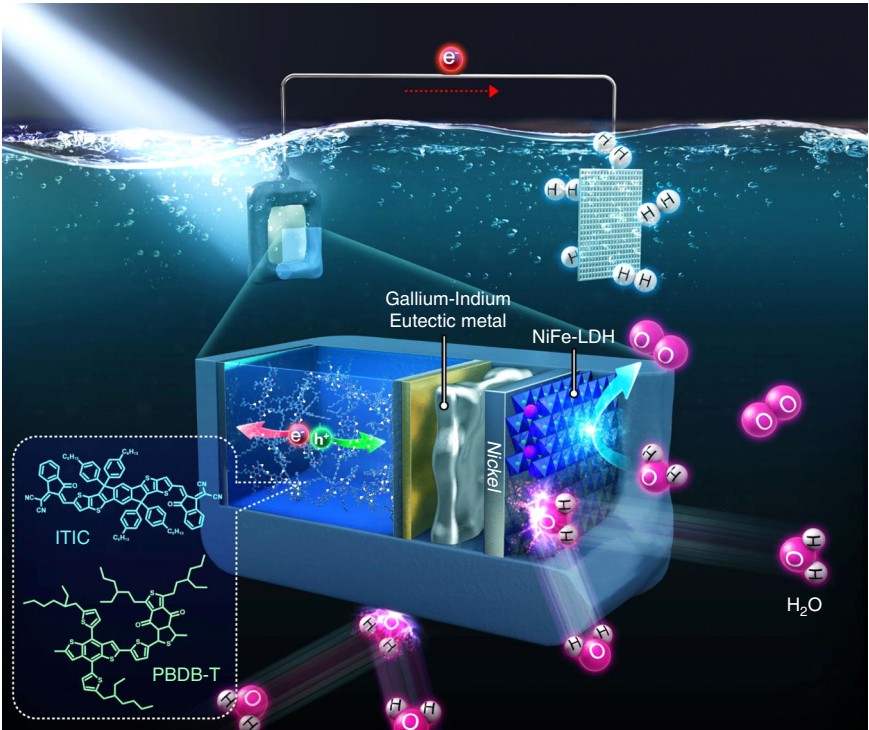

**Fig. 1 Overall schematic of organic-photoactive-layer-based photoanode configuration.** In this scheme, nickel iron layered double hydroxides (NiFe-LDHs), nickel foils, and Gallium Indium eutectic not only protect the organic-photoactive-layer from water but also help the efficient charge transport and separation for the high-performance photoelectrochemical water splitting.

substrate with a roughness of approximately 1.14 nm (Fig. 2a and Supplementary Fig. 4). The photovoltaic performances of the pristine PBDB-T:ITIC-based OPVs are measured under AM 1.5 G solar simulated light (100 mW cm$^{-2}$). The current density–voltage (*J–V*) curve and the corresponding external quantum efficiency (EQE) are shown in Fig. 2b, c, respectively. The device exhibits a power conversion efficiency of 9.44% with 16.4 mA cm$^{-2}$ short-circuit current density ($J_{SC}$), 0.902 V open-circuit voltage ($V_{OC}$), and 63.6% fill factor. The EQE is more than 70% in both 550 and 580 nm regions, which agrees well with the $J_{SC}$ obtained from the *J–V* curve within 5% mismatch.

After passivation of the non-fullerene-based OPVs by using the GaIn eutectic and Ni foil (thickness of 100 μm), the device is sealed with epoxy for fixing the device structure and protecting it from water intrusion. The PEC oxygen evolution performances of the organic-photoactive-layer-based photoanode (denoted as Ni/eu@nfOP) are evaluated through a three-electrode system with the organic photoanode, a Hg/HgO electrode, and a platinum mesh as the working, reference, and counter electrodes, respectively, in a 1 M NaOH (pH 13.6) electrolyte under AM 1.5 G illumination (100 mW cm$^{-2}$) (Fig. 2d). In this alkaline electrolyte, the surface of the Ni metal is transformed to Ni(OH)$_2$/NiOOH, which is an active OER catalyst[37–39]. The photocurrent density is 14.7 mA cm$^{-2}$ at 1.23 V vs. RHE with an onset potential of 0.63 V vs. RHE (the area of all the photoelectrodes is 0.5 cm$^2$). Without these passivation layers, the onset potential of the pristine organic-photoactive-layer-based photoanode (denoted as nfOP) is 1.20 V vs. RHE, which is much more positive than that of Ni/eu@nfOP by more than 0.57 V and the photocurrent density is negligible at 1.23 V vs. RHE. In addition, nfOP is found to be destroyed through contact with the electrolyte for PEC measurement. We cannot observe the photocurrents from the organic-photoactive-layer-based photo-anode with a Ni passivation layer without GaIn eutectic (denoted as Ni@nfOP). The series of results indicate that a Ni foil is an

essential component to passivate OPVs from water and GaIn eutectic is a key element for efficient charge transport.

**Oxygen evolution catalyst optimization.** As mentioned above, Ni works well as an OER catalyst in an alkaline electrolyte. However, to maximize the charge-separation efficiency, we grow NiFe-LDHs that are non-noble-metal-based materials reported as high-performance OER catalysts[40]. LDHs are an important class of ionic lamellar solids that consist of positively charged brucite-like host hydroxide layers and interlayer anions, with unique features such as accommodation of a wide range of cations and controllable cation ratios[41,42]. In addition, they exhibit a characteristic of interspersion of cations on an atomic scale, and thus, no cation segregation occurs within a hydroxide layer[43,44]. In particular, NiFe-LDHs have been reported as one of the most effective OER catalysts in alkaline environments[45–50]. Although further investigations are still needed to identify the active sites, the synergistic interactions between Ni and Fe can indeed enhance the catalytic activity compared to the individual Ni and Fe components[45,51–53]. Because NiFe-LDHs can be readily formed on metal foils[36,53], in this study, we grow NiFe-LDH catalysts directly on a Ni foil used as a passivation layer by a one-step hydrothermal method (see details in the "Experimental" section). Therefore, the NiFe-LDH/Ni foil plays dual roles as both a passivation material and an efficient catalyst.

Scanning electron microscopy (SEM) images (Fig. 3a, b) show the NiFe-LDHs formed on the Ni foil exhibit a 3D porous structure. The NiFe-LDH plates with lateral sizes of several hundred nanometers are interconnected. The X-ray diffraction (XRD) pattern (Fig. 3c) shows the reflection peaks of (003), (006), (012), (015), and (018) planes, matched with those of typical LDHs (JCPDS: 00-014-0191) in addition to the substrate Ni peaks (JCPDS: 00-001-1260), indicating that NiFe-LDHs are successfully synthesized on the Ni foil. The energy-dispersive X-ray

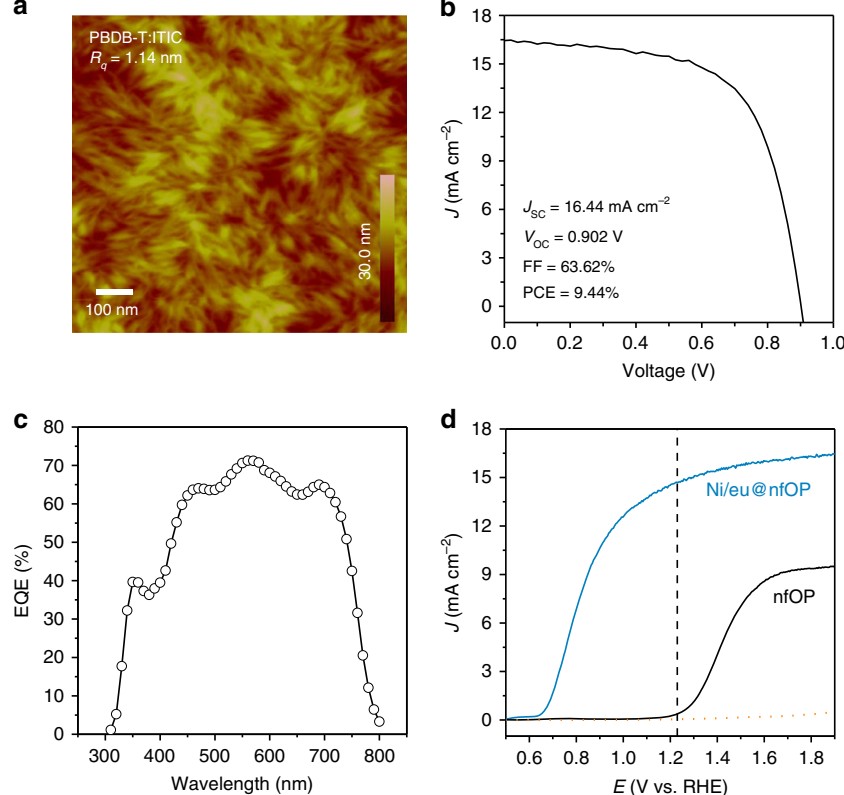

**Fig. 2 Photovoltaic of PBDB-T:ITIC-based OPVs and performance of this organic-photoactive-layer-based photoanode. a** AFM image of the photoactive layer. **b** J–V curve of PBDB-T:ITIC-based OPVs under AM 1.5 G solar simulator (100 mW cm$^{-2}$). **c** EQE for PBDB-T:ITIC OPV. **d** Comparison of current density–potential curve on pristine PBDB-T:ITIC organic photoanode with and without 100-μm-thick Ni foil in 1 M NaOH electrolyte (pH 13.6) under AM 1.5 G solar simulator (100 mW cm$^{-2}$).

spectroscopy (EDS) analysis validates the presence of Ni, Fe, and O in the structures (Fig. 3d).

**Performance of organic-photoactive-layer-based photoanodes.**
We first measure and compare the OER electrocatalytic performances of the devices with the bare Ni foil and NiFe-LDH-modified Ni foil electrodes to further verify the LDH formation. The potentials required for the current density of 10 mA cm$^{-2}$ decrease from 1.60 to 1.50 V vs. RHE upon LDH modification (Supplementary Fig. 5). The results indicate that NiFe-LDHs are successfully grown on the Ni foil and the hole-transfer characteristic was improved for efficient water splitting. Next, we use a NiFe-LDH/Ni foil for passivation of organic photoactive layers with GaIn eutectic to fabricate the device (denoted as LDH/Ni/eu@nfOP). The current density of LDH/Ni/eu@nfOP is 15.1 and 15.6 mA cm$^{-2}$ at 1.23 and 1.33 V vs. RHE (Fig. 3e), respectively, which are very close to the $J_{sc}$ of the pristine PBDB-T:ITIC (16.4 mA cm$^{-2}$) (Fig. 2b). These values are more than two times higher than the best photocurrent density value of metal oxide photocatalysts under the same conditions[13,14,16]. The onset potential of LDH/Ni/eu@nfOP (0.55 V vs. RHE) is shifted to a negative direction by 0.12 and 0.65 V compared to those of Ni/eu@nfOP and nfOP, respectively (Figs. 3e and 2d). This value is comparable to those of metal oxides and far more negative than those of silicon-based photoanodes (0.9–1.2 V)[54]. The photovoltage estimated by open-circuit measurement under dark and light conditions is 0.76 V (Supplementary Fig. 6), which is already far higher than that of the silicon photoelectrode (0.4–0.6 V)[34]. It can be further enhanced by using an organic photoactive layer possessing higher $V_{OC}$ and optimizing the passivation processes. The $\eta_{half\text{-}STH}$, which exhibits the overall performance of

photoelectrodes where both the photocurrent density and onset potential (photovoltage) are considered, is expressed as

$$\eta_{half-STH} = \frac{\left(1.23 - V_{app}\right) \times J_{op}}{P_{in}} \times 100\%, \quad (1)$$

where $V_{app}$ and $J_{op}$ are the applied external bias voltage vs. counter electrode (Pt) in a two-electrode configuration and photocurrent density of a photoelectrode under 1 sun illumination, respectively. $P_{in}$ is the power density of the illuminating light (100 mW cm$^{-2}$), and the power density of the incident sunlight is AM 1.5 G illumination[55] in this study. The $\eta_{half-STH}$ values of LDH/Ni/eu@nfOP is 4.33% at 0.82 V vs. Pt (Fig. 3f and Supplementary Fig. 7), which is the highest value among those of all reported photoanodes, including metal–oxide-, metal–nitride-, silicon-, and other organic semiconductor-based materials, to the best of our knowledge (Supplementary Table 1).

Chronoamperometry (J–t) measurements are conducted under AM 1.5 G illumination at 1.3 V vs. RHE to test the photoanode stabilities (Fig. 3g). Without this passivation, the bare organic photoanode (nfOP) loses its activity within a few minutes. We also find that all organic materials are damaged after the J–t measurement (Supplementary Fig. 8). With Ni foil and GaIn eutectic passivation, 66.5% of the initial photocurrent density of Ni/eu@nfOP is retained after 10 h of light illumination. On the other hand, 90% of the initial performance of LDH/Ni/eu@nfOP is retained, indicating that the stability is significantly improved using NiFe-LDH catalysts. The Faradaic efficiencies for PEC H$_2$ and O$_2$ evolution of LDH/Ni/eu@nfOP at 0.8 V and 1.3 V vs. RHE are near 100% and their H$_2$/O$_2$ molar ratio are also closed to 2 (Fig. 3h, Supplementary Figs. 9 and 10). Moreover, we measure

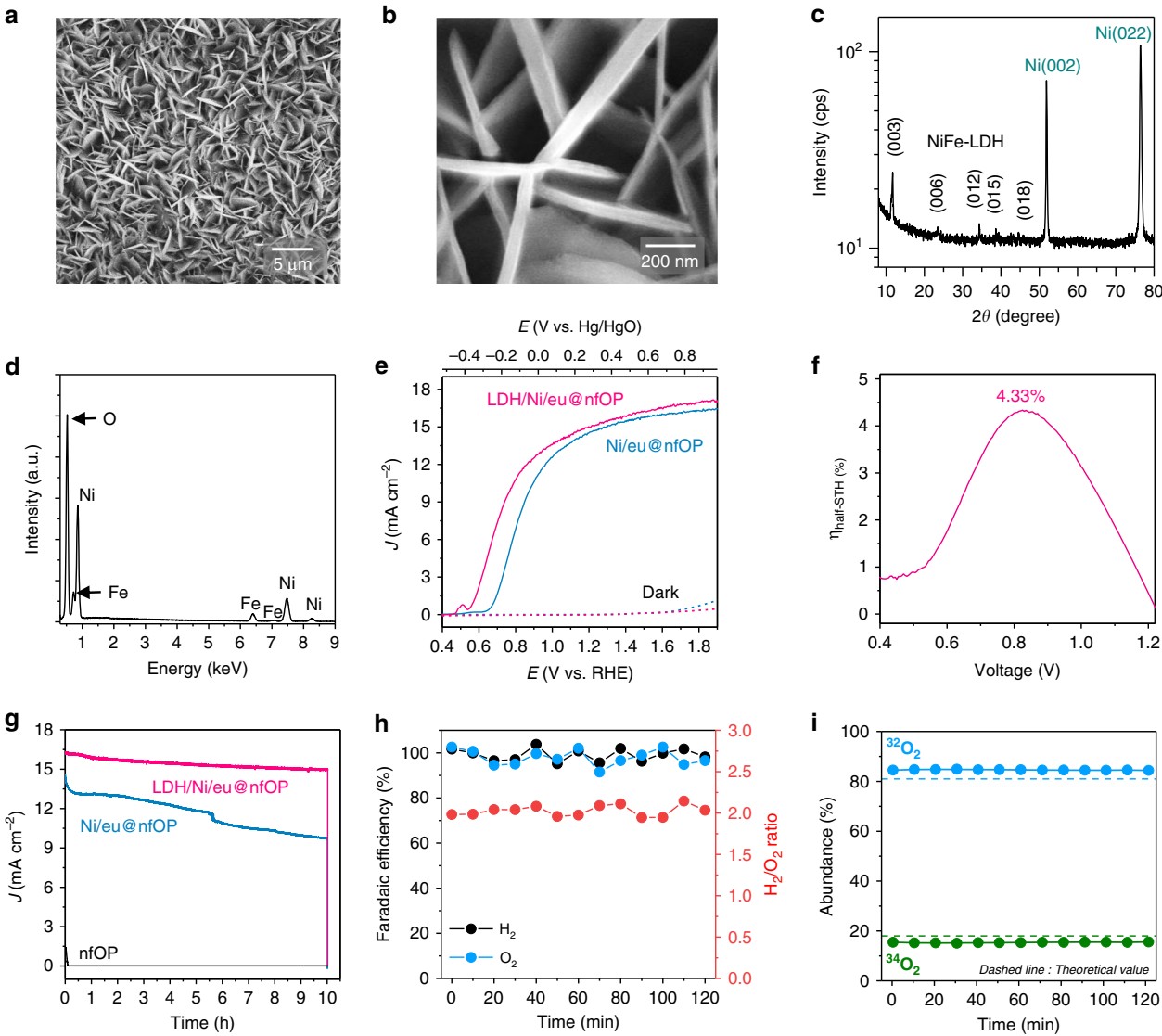

**Fig. 3 Characterization of NiFe-LDH catalysts and PEC performances of passivated organic-photoactive-layer-based photoanodes. a, b** SEM images. **c** XRD $\theta$–$2\theta$ scan pattern. **d** EDS spectrum of NiFe-LDH structures on Ni foil. **e** Comparison of current density–potential curves of Ni/eu@nfOP and LDH/Ni/eu@nfOP in 1 M NaOH electrolyte (pH 13.6) under AM 1.5 G illumination (100 mW cm$^{-2}$). **f** $\eta_{half-STH}$ of LDH/Ni/eu@nfOP measured in a two-electrode configuration. **g** Chronoamperometry measurement and comparison of nfOP, Ni/eu@nfOP, and LDH/Ni/eu@nfOP in 1 M NaOH electrolyte (pH 13.6) at 1.3 V vs. RHE under AM 1.5 G illumination (100 mW cm$^{-2}$). **h** Faradaic efficiencies for photoelectrochemical H$_2$ and O$_2$ production and the H$_2$/O$_2$ ratio at 15.4 mA cm$^{-2}$ (approximately 1.3 V vs. RHE) using LDH/Ni/eu@nfOP. **i** Abundance of $^{32}$O$_2$ and $^{34}$O$_2$ during photoelectrochemical water splitting in $^{18}$O-labeled water (H$_2$$^{18}$O, 10 vol%) added 1 M NaOH solution using LDH/Ni/eu@nfOP.

the ratio of labeled O$_2$ evolution using $^{18}$O-labeled H$_2$O (10%)-added 1 M NaOH electrolyte (Supplementary Fig. 11 and Fig. 3i). The abundance of $^{34}$O$_2$ is 15.4% ± 0.1%, which is about 85% of the theoretical value [Abundance of $^{34}$O$_2$ ($^{16}$O$^{18}$O and $^{18}$O$^{16}$O) = 0.9 × 0.1 × 2 × 100 = 18%]. One of the important reasons for the difference between actual the abundance of $^{34}$O$_2$ and its theoretical value is the kinetic isotope effect in which the reaction rate of $^{18}$O-labeled H$_2$O oxidation reaction is slower than that of the H$_2$O ($^{16}$O) oxidation[56]. These results above of Faradaic efficiencies and $^{18}$O-labeled water experiments indicate that the high and stable photocurrent is mainly utilized for water splitting reactions rather than for other side reactions. Additionally, it needs to be noted that LDH/Ni/eu@nfOP is more stable than that of the fullerene-based organic device (denoted as LDH/Ni/eu/fOP) (Supplementary Fig. 12; we use the BHJ of PBDB-T donor and [6,6]-phenyl C$_{71}$ butyric acid methyl ester (PC$_{71}$BM) acceptor for the fullerene-based organic material (fOP); see Supplementary

Figs. 13 and 14 for detailed information on fOP, such as AFM images and the performance).

**Origin of destabilization of organic-photoactive-layer-based photoanodes.** As suggested in a recent study[57], there are three major factors for destabilizing a photoelectrode: decrease in chemical stability, photocorrosion by accumulated surface-reaching holes, and decrease in photostability of organic photoactive layers themselves. First, to reveal the effect of chemical stability of LDH/Ni/eu@nfOP on the PEC performance, we measure $J$–$E$ performances for five consecutive days after the immersion of the electrode in 1 M NaOH electrolyte for a day without light illumination. There is not much change in its performance, indicating that water permeation is effectually prevented by LDH, Ni, and GaIn eutectic layers (Fig. 4a). Second, stability tests are conducted with and without sulfite, which acts

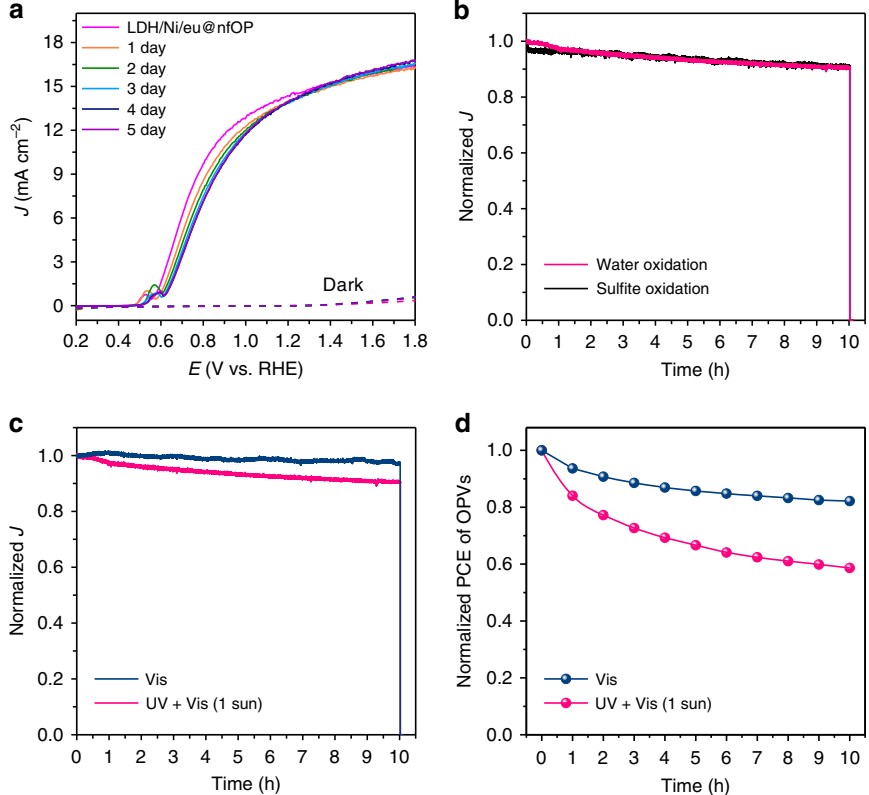

**Fig. 4 Chemical stability test, surface hole-induced photodegradation, and photostability test of the organic photoactive layers of LDH/Ni/eu@nfOP. a** Comparison of current density–potential curves of LDH/Ni/eu@nfOP for five consecutive days immersed in 1 M NaOH electrolyte (pH 13.6) under AM 1.5 G illumination (100 mW cm$^{-2}$) to check the chemical stability. **b** Comparison of normalized current density–time curves of LDH/Ni/eu@nfOP for water oxidation and sulfite oxidation at 1.3 V vs. RHE. For sulfite oxidation, 0.4 M Na$_2$SO$_3$ is added. **c** Comparison of normalized current density–time curves of LDH/Ni/eu@nfOP under AM 1.5 G illumination (UV + Vis, 100 mW cm$^{-2}$) and visible light (with 420 nm cut-off filter, 100 mW cm$^{-2}$). **d** Comparison of normalized efficiency of PBDB-T:ITIC-based OPVs under AM 1.5 G illumination (UV + Vis) and visible light (with 420 nm cut-off filter).

as a hole scavenger. The shapes of the normalized stability curves with and without sulfite are almost the same for 10 h (Fig. 4b), which indicates that holes are not accumulated and are extracted well for water oxidation on LDH/Ni/eu@nfOP. Their J–E characteristics in the solution with and without sulfite are very similar with the same onset potentials and photocurrent densities, further verifying that the NiFe-LDH OER catalysts enable very efficient surface-charge separation (Supplementary Fig. 15). Without NiFe-LDH formation, the stability of Ni/eu@nfOP is much worse than that of LDH/Ni/eu@nfOP (Fig. 3g). However, when there exists sulfite in the solution, the stability of Ni/eu@nfOP is significantly improved with the increased surface charge separation efficiency (Supplementary Figs. 16 and 17). To rule out the possibilities of other reactions such as surface oxidation of LDH/Ni/eu@nfOP the during water splitting reactions, we performed X-ray photoelectron spectroscopy (XPS) and scanning electron microscopy (SEM) before and after the PEC water splitting reaction. (Supplementary Fig. 18). The high resolution Ni 2p and Fe 2p spectra regions show splitting into 2p$_{3/2}$ and 2p$_{1/2}$ doublets due to spin–orbit coupling[50]. The binding energy peaks of Ni 2p$_{3/2}$ and 2p$_{1/2}$ are located at 855.1 and 872.6 eV (Supplementary Fig. 18a, b), respectively, which fit Ni$^{2+}$[52,53,58,59]. Furthermore, the XPS peaks at binding energies of 712.3 and 724.1 eV correspond to Fe 2p$_{3/2}$ and 2p$_{1/2}$, indicating the +3 oxidation state of Fe[52,58,59] (Supplementary Fig. 18c, d). When comparing of XPS spectra of LDH/Ni/eu@nfOP before and after the 10 h stability test, no appreciable change in binding energy is shown. Also, no difference in surface morphology of LDH/Ni/eu@nfOP between

before and after the 10 h stability test is verified by SEM observations (Supplementary Fig. 19). These results show that NiFe-LDHs play a crucial role in surface-charge separation, and without them, the accumulated holes can accelerate photocorrosion of the organic photoactive layer.

Although the passivation strategy by using LDH, Ni foil, and GaIn eutectic is effectual to preserve the chemical stability and prevent photocorrosion by accumulated surface charges, there is still a gradual decrease in photocurrent density during the 10 h operation (Fig. 3g). This suggests that the main reason for the destabilization is the low photostability of organic photoactive layers themselves. It is known that organic photoabsorbers are not stable, particularly under ultraviolet (UV) light rather than visible (Vis) light[60–66]. We can also observe that there was negligible decrease, and 15% of decrease in the absorbance of organic photoactive layers after 10 hours of Vis light irradiation (420 nm cut-off filter, 100 mW cm$^{-2}$) and AM 1.5 G illumination (UV + Vis, 100 mW cm$^{-2}$), respectively (Supplementary Fig. 20). To assess photostabilities of our passivated organic materials, we perform a stability test under Vis light (Fig. 4c and Supplementary Fig. 21). Under Vis-light irradiation, almost 98% of the initial photocurrent density is retained, indicating that photons, especially in the UV region, are the main reason for the stability loss. For further investigation, we measure the J–V characteristics of OPVs each hour during 10 h of light irradiation (Fig. 4d and Supplementary Fig. 22a–c). Their performance deteriorates more rapidly under UV light than that under visible light, which is similar to the J–t characteristics of LDH/Ni/eu@nfOP during PEC measurements.

Therefore, the main reason for the destabilization of our passivated organic-photoactive-layer-based photoanode is not water permeation or surface-charge accumulation but photodegradation of organic materials under light irradiation, particularly UV light.

## Discussion

We demonstrated strategies to significantly enhance the performance and stability of organic-photoactive-layer-based photoanodes by passivation using Ni foils, GaIn eutectic, and NiFe-LDHs. The photocurrent density of LDH/Ni/eu@nfOP was 15.1 mA cm$^{-2}$ at 1.23 V vs. RHE under AM 1.5 G illumination, which is far higher than that of metal oxide photoanodes, and the obtained onset potential of 0.55 V vs. RHE was much more negative than that of Si photoanodes. Based on these high photocurrent and low onset potential, 4.33% $\eta_{\text{half-STH}}$ was recorded. With this passivation strategy, more than 90% of the initial performance of LDH/Ni/eu@nfOP was retained during the 10 h $J$–$t$ measurements, whereas the organic-photoactive-layer-based photoanode without passivation (nfOP) lost its activity within a few minutes. We also revealed that water permeation and photocorrosion by accumulated surface charges can be effectually prevented by passivation using Ni foils with GaIn eutectic and NiFe-LDHs, respectively. The main reason for the activity loss of LDH/Ni/eu@nfOP was low photostability of the organic photoactive layers under irradiation comprising UV components (100 mW cm$^{-2}$), as more than 98% of the initial LDH/Ni/eu@nfOP was retained under Vis light. This study provided a method to effectively passivate an organic semiconductor in water. Therefore, we believe that if highly photostable organic photoactive layers are developed and applied following this strategy, high-performance organic-photoactive-layer-based photoanodes can be achieved with high stabilities compared to inorganic material. Finally, by precisely controlling the conduction and valence band edge positions of organic semiconductors to meet the hydrogen and oxygen evolution potentials, unbiased water splitting would be possible, which is considered as the most ideal form of PEC water splitting.

## Methods

**Preparation of OPV devices**. The patterned ITO glass substrates (15 $\Omega^{-1}$, 1.5 × 1.5 cm$^2$) were cleaned by ultrasonic treatment in detergent, distilled water, acetone, and isopropyl alcohol, and then dried in an oven overnight at 70 °C. The ZnO precursor solution was prepared by dissolving 0.2 g of zinc acetate dihydrate (Aldrich, 99.999%) and 0.055 ml of ethanolamine (Aldrich, 98%) in 2 ml of 2-metoxyethanol (Aldrich, 99.8%). The cleaned ITOs were treated with oxygen plasma for 5 min and the ZnO precursor was spin-coated at 3000 r.p.m. onto the ITO surface. After being baked at 200 °C for 60 min in air, the ZnO-coated substrates were transferred into a nitrogen-filled glove box. The mixed blend solutions of PBDB-T:ITIC (1:1 w/w) and PBDB-T:PC$_{71}$BM (1:1 w/w) with donor concentration 10 mg ml$^{-1}$ in chlorobenzene were prepared. For optimal morphologies, 1,8-diiodooctane additive was added in v/v ratios of 3 vol% in case of a blend with PC$_{71}$BM and 0.5 vol% in case of ITIC blends. Thermal annealing treatments at 100 and 150 °C for 10 min were performed for the blend with PC$_{71}$BM and that with ITIC, respectively. Finally, 10-nm-thick MoO$_3$ and 100-nm-thick gold films (active area: 0.5 cm$^2$) were thermally evaporated under vacuum (<0.5 × 10$^{-5}$ Pa).

**Fabrication of NiFe-LDHs on Ni foil and organic photoanode**. NiFe-LDHs were synthesized using a hydrothermal method, where 0.3 g Ni(NO$_3$)$_2$·6H$_2$O (Aldrich, 99.9%), 0.36 g Fe(NO$_3$)$_3$·9H$_2$O (Aldrich, 98%), and 0.3 g urea (Aldrich, 99%) were dissolved in 80 ml deionized water. Nickel foil (Alfa Aesar, 99.5%, 100 μm thickness) was cleaned by ultrasonication in acetone, isopropyl alcohol, and ethanol for 3 min each. The aqueous solution and Nickel foil were transferred to a 100 ml Teflon-lined autoclave, which was placed at 120 °C for 12 h and then cooled to room temperature. The foil was washed with deionized water and dried at 60 °C overnight. After connecting a copper wire to the prepared OPVs with silver paste and epoxy, a NiFe-LDH/Ni foil was loaded on the electrode with GaIn eutectic (Aldrich, 99.99%) between them. Finally, an epoxy bond was applied to fix and encapsulate the electrode and dried at room temperature overnight (active area: 0.5 cm$^2$).

**Characterization**. The $J$–$V$ characteristics were recorded using a Keithley 2400 source under the illumination of an AM 1.5 G solar simulator with an intensity of 100 mW cm$^{-2}$. The active area of each sample was 0.5 cm$^2$. The EQE measurements were conducted using Model QEX7 by PV measurements Inc.

(Boulder, Colorado) in ambient air. AFM was collected using a Bruker Dimension Icon Atomic Force Microscope with an RTESP-150 probe in the standard tapping mode. The optical density of PBDB-T:ITIC layer was measured using a UV-2600 UV/Vis spectrophotometer (Shimadzu). Crystalline natures of NiFe-LDHs were examined by XRD patterns acquired using a Bruker AXS D8 Advance X-ray diffractometer equipped with Cu Kα radiation ($\lambda$ = 1.5406 Å). Scans were collected between the $2\theta$ range of 8° and 80° with a step size of 0.02°. The morphology and elemental analysis of NiFe-LDHs were investigated by field-emission SEM and energy-dispersive X-ray spectrometry (EDS), which were performed using a Hitachi High-Technologies S-4800 cold field-emission scanning electron microscope. The accelerating voltages for SEM imaging and EDS were 15 kV.

**PEC measurements**. In general, the three-electrode system was used in PEC measurements with a Hg/HgO reference electrode (RE-61AP, ALS), a Pt wire counter electrode, and an Ivium-n-Stat single-channel potentiostat. The linear sweep voltammetry of Ni/eu@nfOP, LDH/Ni/eu@nfOP, and nfOP was measured in a 1 M NaOH (Alfa Aesar, 98%) electrolyte (pH 13.6) from 0.123 to 1.923 V vs. RHE (scan rate = 1 mV s$^{-1}$) with Ar gas purging. The size of the light absorber of the organic photoanodes was 0.5 cm$^2$, and the light was illuminated with a 300 W Xe arc lamp (Newport, 66902) with an air mass 1.5 global (AM 1.5 G) filter, collimating lens, and infrared filter (water). The potential conversion from the measured potential vs. Hg/HgO reference electrode to the potential vs. RHE was conducted by the equation below

$$E(\text{vs.RHE}) = E(\text{vs.Hg/HgO}) + 0.0592\,\text{V} \times \text{pH} + E_{\text{Hg/Hgo}}(\text{reference})$$

$$\left( E_{\text{Hg/Hgo}}(\text{reference}) = 0.118\,\text{V vs.NHE at 25}\,°\text{C} \right). \tag{2}$$

The photocurrent measurements for sulfite oxidation of Ni/eu@nfOP and LDH/Ni/eu@nfOP were conducted in a 100 ml of 1 M NaOH electrolyte, which contained 0.4 M Na$_2$SO$_3$ (Aldrich, 98%). The same electrolyte condition was used in the 10 h long-term chronoamperometry measurement. The chronoamperometry measurements of Ni/eu@nfOP and LDH/Ni/eu@nfOP were performed at 1.3 V vs. RHE and the same condition was used in each water oxidation and sulfite oxidation current density–time measurement. H$_2$ and O$_2$ gas detection was also carried out at 15.4 mA cm$^{-2}$ (approximately 1.3 V vs. RHE) in a fully sealed reactor. The cell was purged for 30 min with Ar, and the amount of evolved H$_2$ and O$_2$ gases was measured by gas chromatography (YL Instrument, 6500GC system) with mass flow controller (Brooks, 5850E). In addition, O$_2$ evolution from the water was confirmed by using 10 vol% $^{18}$O-labeled water (Huayi, 98%)-added in 1 M NaOH, and Ar gas was purged. Mass spectroscopy (RealTek, RGA) was utilized to measure the ratio of $^{18}$O-labeled O$_2$, $^{32}$O$_2$, and $^{34}$O$_2$.

## Data availability
Source data are provided with this paper.

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

## Acknowledgements

This research was supported by the National Research Foundation (NRF) funded by the Ministry of Science and ICT (2019R1F1A1064245, 2019M1A2A2065612, 2018R1C1B6002342, 2017M1A2A2087630, 2017R1D1A1B03035450, and 2018R1A2A1A05077194), Wearable Platform Materials Technology Center (WMC; 2016R1A5A1009926) funded by the NRF Grant by the Korean Government (MSIT),

and the 2018 Research Fund (1.190013.01) of UNIST (Ulsan National Institute of Science and Technology).

## Author contributions

J.-W.J., C.Y., and S.C. proposed, designed, and directed the research. J.M.Y. and J.-W.J. conceived the concept of stabilizing organic photoanodes. J.L. prepared the organic photovoltaics and measured their performances with the help of J.O., S.M.L., and M.J. Y. S.K. synthesized NiFe-LDHs on Ni foils and Y.S.K. and J.S. characterized the NiFe-LDHs. J.M.Y. fabricated organic photoanodes and measured their performances. Y.K. and J.H.K. performed and analyzed $^{18}$O-labeled water experiment. J.M.Y., J.L., Y.S.K., S.C., C.Y., and J.-W.J. co-wrote the paper. All authors read and commented on the paper.

## Competing interests

The authors declare no competing interests.
