## [Peer Review File · Nature Communications]

Reviewers' comments:

Reviewer #1 (Remarks to the Author):

The paper deals with a hot research topic of solar energy conversion to chemical energy. The current density of 15.4 mA/cm² is unusually high in the research field. If it is really due to the water splitting to H₂ and O₂, the paper is worth a publication in Nature Communications. However, the reviewer doubts two points as below and does not recommend to publish at the present manuscript.

1) Oxygen is not measured in the research. Quantitative and 100 % faraday efficiency must be evaluated, if the authors claim their conclusions.

2) The control experiments are not sufficient to deny other oxidation processes.

The authors shows unbelievably high photoanodic current density, especially with GaIn, while the control experiment only with GaIn on the organic layer is not shown. The authors should recognize that there are many possibilities to exhibit photoanodic current, for example metal oxidation etc.

Another reason to doubt the water splitting.

3) The onset potential is not well matched with the theoretical potential.

The potential gain of the OPV is 0.9 V as shown in Fig 1b, and the onset potentials (V_a) for anodic current for NiFe-LDH and Ni Foil were 1.4 V and 1.6 V, respectively, vs RHE (Supplementary Figure 4). From these values, The onset potential for the photoanodic current (V_b) maybe 0.5 V and 0.7 V, respectively, vs RHE. (V_b = V_a-0.9 V) The experimental results are 0.45 V and 0.65 V vs RHE (Figure 2e), which are 0.05V more negative than the calculated values, and means more gain from the calculations. Why the more gain was obtained??

The comparison of current density between Supplementary Figure 4 and Figure 2e gives me another similar conflict to enhance current density.

4) Relating to the calculation, The authors chose vs RHE as x axis while the actual experiment is Hg/HgO standard. The reviewer recommends the authors to show both vs RHE and vs Hg/HgO, because it depends on pH.

A comment

5) There are several attempts to protect the OPV active layer by efficient catalysts. The authors should refer such papers. There are review papers on the field.

L. Steier, S. Holliday, J. Mater. Chem. A, 2018, 6, 21809.

Sultan Otep, Tsuyoshi Michinobu, and Qichun Zhang, Sol. RRL 2019, 1900395.

Reviewer #2 (Remarks to the Author):

Je Min Yu et al. report on a (PBDB-T:ITIC-OPV)/(eutectic metal)/(Ni-LDH catalyst) photoanode claiming stabilized record solar-to-hydrogen efficiency. However, without a gas measurement confirming oxygen evolution at the photoanode with the expected stoichiometric H₂:O₂ molar ratios, the claims in this manuscript are unsupported. Measurement of the gas evolved at the counter electrode (in this case hydrogen) is not sufficient to claim Faradaic efficiency for the water splitting reaction, especially not in view of the observed electrode degradation in Fig. S7. Also the shape of the J-V curve in Fig. 3a might hint to a side reaction such as corrosion occurring at the photoanode. The authors would need to explain the shoulder in the J-V curve, ideally showing O₂ evolution at potentials before and after the shoulder is passed (i.e. 0.55 and 1 V vs RHE)

While the use of the eutectic is not a new concept, the PBDB-T:ITIC bulk heterojunction is novel in the field of polymer photoelectrochemical cells (anodes or cathodes). It is striking though, that the authors entirely omit citations of previous works on polymer photoelectrochemical cells. Admittedly, only few works exist demonstrating polymer photoanodes (i.e. BBL: Borno et al., J Am Chem Soc. 2015, 137, 15338–15341), but there are many works on polymer photocathodes including the works on organic photovoltaics used in electrolyte solutions (i.e. Bourgeteau et al., ACS Appl. Mater.

Interfaces 2015, 7, 16395-16403) and others cited in recent reviews (Steier, Holiday, J. Mater. Chem. A, 2018, 6, 21809-21826; Bellani, Antognazza, Bonaccorso, Adv. Mater. 2019, 31, 1801446.) I encourage the authors to cite previous works on polymer materials and bulk heterojunctions used in photoelectrochemical water splitting to put their work into context.

Response to Comments of Reviewer #1

General Comment: The paper deals with a hot research topic of solar energy conversion to chemical energy. The current density of 15.4 mA/cm^2 is unusually high in the research field. If it is really due to the water splitting to H_2 and O_2 , the paper is worth a publication in Nature Communications. However, the reviewer doubts two points as below and does not recommend to publish at the present manuscript.

Response: Thank you for the reviewer's comments. We would like to carefully address them one by one.

Comment 1: Oxygen is not measured in the research. Quantitative and 100 % faraday efficiency must be evaluated, if the authors claim their conclusions.

Response: We thank the reviewer for the valuable comment. Faradaic efficiency of O_2 as well as H_2 should be measured to prove that the high photocurrent originally comes from water splitting reaction not from other reactions as reviewer's comment.

We measured the amounts of H_2 and O_2 evolution during the photoelectrochemical water splitting reaction at 15.4 mA cm^{-2} (approximately 1.3 V vs. RHE) using LDH/Ni/eu@nfOP (revised Supplementary Fig. 9). The faradaic efficiencies for H_2 and O_2 evolution were almost 100%, and the H_2/O_2 molar ratio was also closed to 2 indicating that the high photocurrent over 15 mA cm^{-2} is originated from water splitting reactions not from other reactions (revised Fig. 2h).

Revised Supplementary Fig. 9 | The potential–time curve of LDH/Ni/eu@nfOP at 15.4 mA cm^{-2} in 1 M NaOH under AM 1.5G illumination.

Revised Fig. 2h, Faradaic efficiencies for photoelectrochemical H₂ and O₂ production and the H₂/O₂ ratio at 15.4 mA cm⁻² (approximately 1.3 V vs. RHE) using LDH/Ni/eu@nfOP.

Furthermore, we measured the ratio of labelled O₂ (³⁴O₂) evolution in ¹⁸O-labelled H₂O (10%)-added 1 M NaOH solution (revised Fig. 2i). The abundance of ³⁴O₂ was 15.4% ± 0.1% of the total amount of O₂ at each sampling point (revised Supplementary Fig. 10), confirming that O₂ was produced from the water oxidation reaction, not from other O₂ sources. One of the reasons why the amount of ³⁴O₂ is not exactly same as the theoretical value of 18% (Theoretical abundance of ³⁴O₂ (¹⁶O¹⁸O and ¹⁸O¹⁶O) = 0.9 × 0.1 × 2 × 100 = 18%) is that ¹⁸O-labelled H₂O oxidation reaction is slower than the H₂O (¹⁶O) oxidation, which is called the kinetic isotope effect (*Chem. Sci.*, 2014, 5, 1141).

Revised Fig. 2i, Abundance of $^{32}\text{O}_2$ and $^{34}\text{O}_2$ during photoelectrochemical water splitting in ^{18}O -labelled water (H_2^{18}O , 10%) added 1 M NaOH solution using LDH/Ni/eu@nfOP.

Revised Supplementary Fig. 10 | Mass spectra of $^{32}\text{O}_2$ and $^{34}\text{O}_2$ evolution of LDH/Ni/eu@nfOP in ^{18}O -labelled H_2O (10 %) added 1 M NaOH under dark and AM 1.5G illumination.

Based on the above results, we can insist that the photocurrent is mainly used for water oxidation reaction not for other side reactions.

Manuscript text, Page no. 14

~~“The Faradaic efficiency of PEC H₂ evolution of LDH/Ni/eu@nfOP is 100%”~~

→ “The Faradaic efficiencies for PEC H₂ and O₂ evolution of LDH/Ni/eu@nfOP are near 100% and the H₂/O₂ molar ratio is also closed to 2 (Supplementary Fig. 9 and Fig. 2h). Moreover, we measure the ratio of labelled O₂ evolution using ¹⁸O-labelled H₂O (10%)-added 1 M NaOH electrolyte (Fig. 2i). The abundance of ³⁴O₂ is 15.4% ± 0.1%, which is about 85 % of the theoretical value [Abundance of ³⁴O₂ (¹⁶O¹⁸O and ¹⁸O¹⁶O) = 0.9 × 0.1 × 2 × 100 = 18%]. One of the important reasons for the difference between actual the abundance of ³⁴O₂ and its theoretical value is the kinetic isotope effect in which the reaction rate of ¹⁸O-labelled H₂O oxidation reaction is slower than that of the H₂O (¹⁶O) oxidation. These results above of Faradaic efficiencies and ¹⁸O-labelled water experiments indicate that the high and stable photocurrent is mainly utilized for water splitting reactions rather than for other side reactions. Additionally, it needs to be noted that”

Manuscript text, Page no. 22 (Methods)

~~“H₂ gas detection was also carried out at 1.3 V vs. RHE with a three-electrode system in a fully sealed reactor. The amount of evolved H₂ gases was measured by gas chromatography (YL Instrument, 6500GC system) with argon carrier gases injected at 5 sccm rates using a mass flow controller (Brooks, 5850E). In addition, the calibration value was defined using standard mixture gas (SUPELCO, 22561).”~~

→ “H₂ and O₂ gas detection was also carried out at 15.4 mA cm⁻² (approximately 1.3 V vs. RHE) in a fully sealed reactor. The cell was purged for 30 min with Ar, and the amount of evolved H₂ and O₂ gases was measured by gas chromatography (YL Instrument, 6500GC system) with mass flow controller (Brooks, 5850E). Additionally, O₂ evolution from the water was confirmed by using 10 vol. % ¹⁸O-labelled water (Huayi, 98 %) added in 1 M NaOH, and Ar gas was purged. Mass spectroscopy (RealTek, RGA) was utilized to measure the ratio of ¹⁸O-labelled O₂, ³²O₂, and ³⁴O₂.

Revised Fig. 2 | Characterization of NiFe-LDH catalysts and PEC performances of passivated organic-photoactive-layer-based photoanodes. a, b, SEM images. c, XRD θ - 2θ scan pattern. d, EDS spectrum of NiFe-LDH structures on Ni foil. e, Comparison of current density-potential curves of Ni/eu@nfOP and LDH/Ni/eu@nfOP in 1 M NaOH electrolyte (pH 13.6) under AM 1.5G illumination (100 mW cm^{-2}). f, $\eta_{\text{half-STH}}$ of Ni/eu@nfOP and LDH/Ni/eu@nfOP. g, Chronoamperometry measurement and comparison of nfOP, Ni/eu@nfOP, and LDH/Ni/eu@nfOP in 1 M NaOH electrolyte (pH 13.6) at 1.3 V vs. RHE under AM 1.5G illumination (100 mW cm^{-2}). h, Faradaic efficiencies for photoelectrochemical H_2 and O_2 production and the H_2/O_2 ratio at 15.4 mA cm^{-2} (approximately 1.3 V vs. RHE) using LDH/Ni/eu@nfOP. i, Abundance of $^{32}\text{O}_2$ and $^{34}\text{O}_2$ during photoelectrochemical water splitting in ^{18}O -labelled water (H_2^{18}O , 10%) added 1 M NaOH solution using LDH/Ni/eu@nfOP.

Comment 2: The control experiments are not sufficient to deny other oxidation processes. While the control experiment only with GaIn on the organic layer is not shown. The authors should recognize that there are many possibilities to exhibit photoanodic current, for example metal oxidation etc.

Response: Thank you for the valuable comments. First we measured the performance of organic photoanode with only GaIn eutectic metal without Ni foil passivation (denoted as eu@nfOP) in accordance with the reviewer's comment. However, the photocurrent was very low (Review only Fig. 1), because GaIn eutectic metal was immediately detached with the organic photoactive layer from the electrode as shown in Review only Fig. 2. For the better understanding of GaIn eutectic loading, we schematically explained all the photoelectrode fabrication processes with the relevant photograph in the revised Supplementary Fig. 3.

Review only Fig. 1 | J-V curves of of Ni/eu@nfOP, Ni/eu@nfOP and nfOP.

Review only Fig. 2 | Photographic images of eu@nfOP. a, before and b, after the immersion of eu@nfOP in electrolyte.

Revised Supplementary Fig. 3 | Schematic and photographic images of organic photoactive-layer-based-photoanode fabrication process.

Manuscript text, Page no. 6

“The detailed process for the photoelectrode fabrication is schematically illustrated with the relevant photographs in Supplementary Fig. 3.”

To check the possibility of side reactions such as oxidation of photoanode, we performed x-ray photoelectron spectroscopy (XPS) and scanning electron microscopy (SEM) of organic photoanode before and after the photoelectrochemical water splitting reaction. We first performed the comparative analysis of the high resolution XPS spectra of the LDH/Ni/eu@nfOP before and after the 10 h stability test (revised Supplementary Fig. 17). The high resolution Ni 2p and Fe 2p spectrum regions of the LDH materials show splitting into 2p_{3/2} and 2p_{1/2} doublets due to spin-orbit coupling (*Adv. Energy Mater.* 2018, 8, 1703189). The binding energy peaks of Ni 2p_{3/2} and 2p_{1/2} are located at 855.1 eV and 872.6 eV (Supplementary Fig. 17a, b), respectively, which fit Ni²⁺ (*J. Am. Chem. Soc.* 2013, 135, 23, 8452-8455, *Chem. Commun.*, 2014, 50, 6479-6482, *J. phys. Chem. Lett.* 2020, 11, 968-973, *Adv. Energy Mater.* 2015, 5, 13, 1500245). Furthermore, the XPS peaks at a binding energy of 712.3 eV and 724.1 eV correspond to Fe 2p_{3/2} and 2p_{1/2}, indicating the +3 oxidation state of Fe (Supplementary Fig. 17c, d) (*J. Am. Chem. Soc.* 2013, 135, 23, 8452-8455, *J. phys. Chem. Lett.* 2020, 11, 968-973, *Adv. Energy Mater.* 2015, 5, 13, 1500245). When comparing of XPS spectra of LDH/Ni/eu@nfOP before and after the 10 h stability test, no appreciable change in binding energy is shown. Additionally, no difference in surface morphology of LDH/Ni/eu@nfOP before and after the 10 h stability test is confirmed by SEM images (revised Supplementary Fig. 18).

Revised Supplementary Fig. 17 | High-resolution XPS spectrum of NiFe-LDH/Ni/eu@nfOP. a, b, Ni 2p, c, d, Fe 2p in LDH/Ni/eu@nfOP before and after 10 h stability test, respectively.

Revised Supplementary Fig. 18 | SEM images of LDH/Ni/eu@nfOP. a,b, Surface of LDH/Ni/eu@nfOP before and c,d, after 10 h stability test.

“However, when there exists sulfite in the solution, the stability of Ni/eu@nfOP is significantly improved with the increased surface charge separation efficiency (Supplementary Fig. 15 and 16. To rule out the possibilities of other reactions such as surface oxidation of LDH/Ni/eu@nfOP during water splitting reactions, we performed X-ray photoelectron spectroscopy (XPS) and scanning electron microscopy (SEM) before and after the photoelectrochemical water splitting reaction. (Supplementary Fig. 17). The high resolution Ni 2p and Fe 2p spectra regions show splitting into 2p_{3/2} and 2p_{1/2} doublets due to spin-orbit coupling⁴⁵. The binding energy peaks of Ni 2p_{3/2} and 2p_{1/2} are located at 855.1 eV and 872.6 eV (Supplementary Fig. 17a, b), respectively, which fit Ni²⁺^{47,48,53,54}. Furthermore, the XPS peaks at binding energies of 712.3 eV and 724.1 eV correspond to Fe 2p_{3/2} and 2p_{1/2}, indicating the +3 oxidation state of Fe^{47,53,54} (Supplementary Fig. 17c, d). When comparing of XPS spectra of LDH/Ni/eu@nfOP before and after the 10 h stability test, no appreciable change in binding energy is shown. Also, no difference in surface morphology of LDH/Ni/eu@nfOP between before and after the 10 h stability test is verified by SEM observations (Supplementary Fig. 18).”

Furthermore, we measured UV-vis spectrum for organic semiconductor materials (PBDB-T:ITIC-based BHJ) before and after the irradiation under AM 1.5G illumination (UV + Vis) and Vis light irradiation respectively. As shown in revised supplementary Fig 19, during 10 h light irradiation, the 15 % of absorbance of PBDB-T:ITIC-based BHJ layer declines after 10 h AM 1.5G illumination (UV + Vis). On the other hand, there is no difference before and after 10 h only Vis light irradiation indicating that there is no significant photodegradation or oxidation of organic semiconductors.

Revised Supplementary Fig. 19 | Comparison of UV-Vis absorption spectra of PBDB-T:ITIC-based BHJ layer before and after 10 h light irradiation. a, Under irradiation of AM 1.5G illumination (UV + Vis) light. b, Under irradiation of only Vis light.

“We can also observe that there is negligible decrease, and 15% of decrease in the absorbance of organic photoactive layers after 10 hours of Vis light irradiation (420 nm cut-off filter, 100 mW cm⁻²) and AM 1.5G illumination (UV+Vis, 100 mW cm⁻²), respectively (Supplementary Fig. 19).”

Comment 3: The onset potential is not well matched with the theoretical potential. The potential gain of the OPV is 0.9 V as shown in Fig 1b, and the onset potentials (V_a) for anodic current for NiFe-LDH and Ni Foil were 1.4 V and 1.6 V, respectively, vs. RHE (Supplementary Figure 4). From there values, The onset potential for the photoanodic current (V_b) maybe 0.5 V and 0.7 V, respectively, vs RHE. (V_b = V_a-0.9 V) The experimental results are 0.45 V and 0.65 V vs RHE (Figure 2e), which are 0.05V more negative than the calculated values, and means more gain from the calculations. Why the more gain was obtained?? The comparison of current density between Supplementary Fig. 4 and Fig. 2e gives me another similar conflict to enhance current density.

Response: We thank for the reviewer’s critical comments. Due to the redox current of nickel-based oxygen evolution catalysts around the onset potential (*J. Am. Chem. Soc.* 2014, 136, 6744–6753; *J. Am. Chem. Soc.* 2017, 139, 11361–11364), we could not get accurate onset potential for the water splitting. Although we are not able to perfectly remove the surface catalyst’s redox current, we can reduce this redox current to the minimum by decreasing the scan rate from 20 mV s⁻¹ to 1mV s⁻¹ during linear sweep voltammetry measurement. Thus we could get more precise values of onset potential in the revised manuscript (revised Fig. 2e and revised Supplementary Fig. 5).

Revised Fig. 2e, Comparison of current density–potential curves of Ni/eu@nfOP and LDH/Ni/eu@nfOP in 1 M NaOH electrolyte (pH 13.6) under AM 1.5G illumination (100 mW cm⁻²).

Revised Supplementary Fig. 5 | Electrochemical performance of Ni foil and NiFe-LDH-loaded Ni electrode.

The onset potentials of LDH/Ni/eu@nfOP (Vb) and NiFe-LDH-modified Ni foil (Va) were 1.42 V and 0.55 V, respectively indicating that photovoltage ($V_b - V_a$) is 0.87 V. This value is lower than that of OPV (0.9 V). The photovoltage (1.52 V – 0.67 V) obtained from Ni/eu@nfOP is also lower than 0.9 V.

Manuscript text, Page no. 12

~~“The onset potential of LDH/Ni/eu@nfOP (0.47 V vs. RHE) is shifted to a negative direction by 0.16 V and 0.73 V compared to those of Ni/eu@nfOP and nfOP”~~

→ “The onset potential of LDH/Ni/eu@nfOP (0.55 V vs. RHE) is shifted to a negative direction by 0.12 V and 0.65 V compared to those of Ni/eu@nfOP and nfOP, respectively”

Comment 4: Relating to the calculation, The authors chose vs RHE as x axis while the actual experiment is Hg/HgO standard. The reviewer recommends the authors to show both vs RHE and vs Hg/HgO, because it depends on pH.

Response: We appreciate the reviewer’s valuable comment. In accordance with the reviewer’s

comment, we provided both E (V vs. Hg/HgO) and E (V vs. RHE), in the revised manuscript (revised Fig. 2e)

Revised Fig. 2e, Comparison of current density–potential curves of Ni/eu@nfOP and LDH/Ni/eu@nfOP in 1 M NaOH electrolyte (pH 13.6) under AM 1.5G illumination (100 mW cm^{-2}).

Comment 5: There are several attempts to protect the OPV active layer by efficient catalysts. The authors should refer such papers. There are review papers on the field.

L. Steier, S. Holliday, *J. Mater. Chem. A*, 2018, 6, 21809.

Sultan Otep, Tsuyoshi Michinobu, and Qichun Zhang, *Solar RRL* 2019, 1900395.

Response: We thank the reviewer for the valuable comment. In accordance with the reviewer's recommendation, we referred the literatures about other organic based photoelectrochemical cells with relevant explanation in the revised manuscript.

Added references

- 16 Steier, L. & Holliday, S. A bright outlook on organic photoelectrochemical cells for water splitting. *J. Mater. Chem. A* 6, 21809-21826 (2018).
- 17 Bellani, S., Antognazza, M. R. & Bonaccorso, F. Carbon-based photocathode materials for solar hydrogen production. *Adv. Mater.* 31, 1801446 (2019).
- 18 Otep, S., Michinobu, T. & Zhang, Q. Pure organic semiconductor-based photoelectrodes for water splitting. *Solar RRL*, 1900395 (2019).
- 19 Bourgeteau, T. et al. A H₂-evolving photocathode based on direct sensitization of MoS₃ with an organic photovoltaic cell. *Energy Environ. Sci.* 6, 2706-2713 (2013).

- 20 Bourgeteau, T. et al. Enhancing the performances of P3HT:PCBM–MoS₃-based H₂-evolving photocathodes with interfacial layers. *ACS Appl. Mater. Interfaces* 7, 16395-16403 (2015).
- 21 Yao, L. et al. Establishing stability in organic semiconductor photocathodes for solar hydrogen production. *J. Am. Chem. Soc.* 142, 7795-7802 (2020).
- 22 Shirakawa, H., Ikeda, S., Aizawa, M., Yoshitake, J. & Suzuki, S. Polyacetylene film: A new electrode material for photoenergy conversion. *Synthetic Metals* 4, 43-49 (1981).
- 23 Borno, P., Prévot, M. S., Yu, X., Guijarro, N. & Sivula, K. Direct light-driven water oxidation by a ladder-type conjugated polymer photoanode. *J. Am. Chem. Soc.* 137, 15338-15341 (2015).
- 24 Dai, C., Gong, X., Zhu, X., Xue, C. & Liu, B. Molecular modulation of fluorene-dibenzothiophene-S,S-dioxide-based conjugated polymers for enhanced photoelectrochemical water oxidation under visible light. *Mater. Chem. Front.* 2, 2021-2025 (2018).
- 25 Ruan, Q. et al. A nanojunction polymer photoelectrode for efficient charge transport and separation. *Angew. Chem., Int. Ed.* 56, 8221-8225 (2017).
- 26 Peng, G., Volokh, M., Tzadikov, J., Sun, J. & Shalom, M. Carbon nitride/reduced graphene oxide film with enhanced electron diffusion length: An efficient photoelectrochemical cell for hydrogen generation. *Adv. Energy Mater.* 8, 1800566 (2018).

Manuscript text, Page no. 4

~~“In addition, organic semiconductors can be used for both photoanodes and photocathodes by simply converting their order of deposition^{16,17}. Despite these merits, organic semiconductors have not attracted attention as photoactive materials for PEC water splitting due to their low stability in aqueous solutions¹⁸.”~~

→ “Several organic semiconductors were applied for photocathodes, since the first demonstration of photocathodic hydrogen production by Shirakawa et al. with polyacetylene in 1981¹⁹⁻²², and only few organic photoactive-layer-based photoanodes have been reported. For example, Borno et al. and Dai et al. showed that the poly[benzimidazobenzophenanthroline] and fluorine-dibenzothiophene-S,S-dioxide-based conjugated polymer can be utilized as photoanodes for direct solar water oxidation, respectively^{23,24}. Ruan et al. and Peng et al. showed the possibility of usage of carbon nitride materials as photoanodes^{25,26}. However, these organic photoactive-layer-based photoanodes not only exhibited photocurrent densities of only few microampere scales at 1.23 V vs. RHE (~100 $\mu\text{A cm}^{-2}$) but half-cell solar-to-hydrogen conversion efficiencies ($\eta_{\text{half-STH}}$) were also lower than 0.03% so far (Supplementary Table 1 and Supplementary Fig. 1). Furthermore, they lost their performances in a few minutes and even for the case of the recent stable organic photoelectrodes stabilized by TiO₂ layers, only less than 50% initial photocurrent is maintained after 1 hour.¹⁸”

Response to Comments of Reviewer #2

Comment 1: Je Min Yu et al. report on a (PBDB-T:ITIC-OPV)/(eutectic metal)/(Ni-LDH catalyst) photoanode claiming stabilized record solar-to-hydrogen efficiency. However, without a gas measurement confirming oxygen evolution at the photoanode with the expected stoichiometric H₂:O₂ molar ratios, the claims in this manuscript are unsupported. Measurement of the gas evolved at the counter electrode (in this case hydrogen) is not sufficient to claim Faradaic efficiency for the water splitting reaction, especially not in view of the observed electrode degradation in Fig. S7. Also the shape of the J-V curve in Fig. 3a might hint to a side reaction such as corrosion occurring at the photoanode. The authors would need to explain the shoulder in the J-V curve, ideally showing O₂ evolution at potentials before and after the shoulder is passed (i.e. 0.55 and 1 V vs RHE)

Response: We're appreciative of the reviewer's insightful and very important comments. We totally agree that O₂ evolution measurement is needed as well as H₂ during the photoelectrochemical reaction to insist that high photocurrent is utilized for water splitting. We first measured the H₂ and O₂ gas evolution during 2 h operation at 15.4 mA cm⁻² condition (approximately 1.3 V vs. RHE) using of LDH/Ni/eu@nfOP (revised Supplementary Fig. 9). The faradaic efficiencies for PEC H₂ and O₂ evolution of LDH/Ni/eu@nfOP were almost 100% and the H₂/O₂ molar ratio was also closed to 2 (revised Fig. 2h). Furthermore, we measured the labelled O₂ ratio during photoelectrochemical reactions on LDH/Ni/eu@nfOP in ¹⁸O-labelled H₂O (10 %)-added 1 M NaOH electrolyte (revised Fig. 2i). The ratio of ³⁴O₂ is 15.4% ± 0.1%, which is about 85 % of the theoretical value [Abundance of ³⁴O₂ (¹⁶O¹⁸O and ¹⁸O¹⁶O) = 0.9 × 0.1 × 2 × 100 = 18%] (revised Supplementary Fig. 10), confirming that O₂ is produced from the water oxidation reaction. One of the important reasons for the difference between the actual abundance of ³⁴O₂ and its theoretical value is the kinetic isotope effect in which the reaction rate of ¹⁸O-labelled H₂O oxidation reaction is slower than that of the H₂O (¹⁶O) oxidation (Ref: *Chem. Sci.*, 2014, 5, 1141).

Revised Supplementary Fig. 9 | The potential – time curve of LDH/Ni/eu@nfOP in 1M NaOH under AM 1.5G illumination at 15.4 mA cm⁻².

Revised Fig. 2h, Faradaic efficiencies for photoelectrochemical H₂ and O₂ production and the H₂/O₂ ratio at 15.4 mA cm⁻² (approximately 1.3 V vs. RHE) using LDH/Ni/eu@nfOP.

Revised Fig. 2i, Abundance of $^{32}\text{O}_2$ and $^{34}\text{O}_2$ during photoelectrochemical water splitting in ^{18}O -labelled water (H_2^{18}O , 10%) added 1 M NaOH solution using LDH/Ni/eu@nfOP.

Revised Supplementary Fig. 10 | Mass spectra of $^{32}\text{O}_2$ and $^{34}\text{O}_2$ evolution of LDH/Ni/eu@nfOP in ^{18}O -labelled H_2O (10 %) added 1 M NaOH under dark and AM 1.5G illumination.

Following these results above, we can insist that the generated photocurrent is efficiently used for water oxidation reaction not for other reactions when the organic-based-photoactive-layer is well passivated by Ni foils and OER catalyast (LDH/Ni/eu@nfOP). However, without this passivation, there was noticeable degradation of photoanode (nfOP) as shown in Supplementary Figure 8.

Supplementary Fig. 78 | Photographic images of nfOP. a, Back side of pristine nfOP. **b,** Front side of pristine nfOP. **c,** Back side of nfOP. **d,** Front side of nfOP after the PEC performance measurement.

Manuscript text, Page no. 14

~~“The Faradaic efficiency of PEC H₂ evolution of LDH/Ni/eu@nfOP is 100%”~~

~~→ “The Faradaic efficiencies for PEC H₂ and O₂ evolution of LDH/Ni/eu@nfOP are near 100% and the H₂/O₂ molar ratio is also closed to 2 (Supplementary Fig. 9 and Fig. 2h). Moreover, we measure the ratio of labelled O₂ evolution using ¹⁸O-labelled H₂O (10%)-added 1 M NaOH electrolyte (Fig. 2i). The abundance of ³⁴O₂ is 15.4% ± 0.1%, which is about 85 % of the theoretical value [Abundance of ³⁴O₂ (¹⁶O¹⁸O and ¹⁸O¹⁶O) = 0.9 × 0.1 × 2 × 100 = 18%]. One of the important reasons for the difference between actual the abundance of ³⁴O₂ and its theoretical value is the kinetic isotope effect in which the reaction rate of ¹⁸O-labelled H₂O oxidation reaction is slower than that of the H₂O (¹⁶O) oxidation⁵¹. These results above of Faradaic efficiencies and ¹⁸O-labelled water experiments indicate that the high and stable photocurrent is mainly utilized for water splitting reactions rather than for other side reactions. Additionally, it needs to be noted that”~~

Manuscript text, Page no. 22 (Methods)

~~“H₂ gas detection was also carried out at 1.3 V vs. RHE with a three-electrode system in a fully sealed reactor. The amount of evolved H₂ gases was measured by gas chromatography (YL Instrument, 6500GC system) with argon carrier gases injected at 5 sccm rates using a mass flow controller (Brooks, 5850E). In addition, the calibration value was defined using standard mixture gas (SUPELCO, 22561).”~~

~~→ “H₂ and O₂ gas detection was also carried out at 15.4 mA cm⁻² (approximately 1.3 V vs. RHE) in a fully sealed reactor. The cell was purged for 30 min with Ar, and the amount of evolved H₂ and O₂ gases was measured by gas chromatography (YL Instrument, 6500GC system) with mass flow controller (Brooks, 5850E). Additionally, O₂ evolution from the water was confirmed by using 10 vol. % ¹⁸O-labelled water (Huayi, 98 %) added in 1 M NaOH, and Ar gas was purged. Mass spectroscopy (RealTek, RGA) was utilized to measure the ratio of ¹⁸O-~~

Revised Fig. 2 | Characterization of NiFe-LDH catalysts and PEC performances of passivated organic-photoactive-layer-based photoanodes. a, b, SEM images. c, XRD θ - 2θ scan pattern. d, EDS spectrum of NiFe-LDH structures on Ni foil. e, Comparison of current density–potential curves of Ni/eu@nfOP and LDH/Ni/eu@nfOP in 1 M NaOH electrolyte (pH 13.6) under AM 1.5G illumination (100 mW cm^{-2}). f, $\eta_{\text{half-STH}}$ of Ni/eu@nfOP and LDH/Ni/eu@nfOP. g, Chronoamperometry measurement and comparison of nfOP, Ni/eu@nfOP, and LDH/Ni/eu@nfOP in 1 M NaOH electrolyte (pH 13.6) at 1.3 V vs. RHE under AM 1.5G illumination (100 mW cm^{-2}). h, Faradaic efficiencies for photoelectrochemical H_2 and O_2 production and the H_2/O_2 ratio at 15.4 mA cm^{-2} (approximately 1.3 V vs. RHE) using LDH/Ni/eu@nfOP. i, Abundance of $^{32}O_2$ and $^{34}O_2$ during photoelectrochemical water splitting in ^{18}O -labelled water ($H_2^{18}O$, 10%) added 1 M NaOH solution using LDH/Ni/eu@nfOP.

Second, we tried to explain the shoulder peaks in the J - V curves. In the recent literatures the shoulder peak around water oxidation potential is known as the redox current generated by oxidation of the Ni based OER catalyst not related to water oxidation (*J. Am. Chem. Soc.* 2014, 136, 6744–6753; *J. Am. Chem. Soc.* 2017, 139, 11361–11364). To minimize this redox current, we decreased the scan rate from 20 mV s^{-1} to 1 mV s^{-1} , and we found that this oxidation peak was dramatically reduced, although we could not perfectly remove these peaks (Review only Fig. 3).

Review only Fig. 3 | Current density – potential curves of LDH/Ni/eu@nfOP with different scan rate.

In Fig. 3a. we can also observed that these redox currents were dramatically reduced when we reduced the scan rate.

Previous Fig. 3a (scan rate = 20 mV s⁻¹)

Revised Fig. 3a (scan rate = 1 mV s⁻¹)

We also updated other J - V curves after changing the scan rate in other in the revised manuscript (revised Fig 2e,f, and revised Supplementary Fig. 5, 14 and 16)

Revised Fig. 2e, Comparison of current density–potential curves of Ni/eu@nfOP and LDH/Ni/eu@nfOP in 1 M NaOH electrolyte (pH 13.6) under AM 1.5G illumination (100 mW cm^{-2}). **f**, $\eta_{\text{half-STH}}$ of Ni/eu@nfOP and LDH/Ni/eu@nfOP.

Revised Supplementary Fig. 5 | Electrochemical performance of Ni foil and NiFe-LDH-loaded Ni electrode.

Revised Supplementary Fig. 14 | Comparison of current density–potential curves of LDH/Ni/eu@nfOP with and without sulfite under AM 1.5G illumination (100 mW cm⁻²). For the sulfite oxidation, we added 0.4 M Na₂SO₃ to 1 M NaOH electrolyte.

Revised Supplementary Fig. 16 | Comparison of surface charge separation efficiency of Ni/eu@nfOP and LDH/Ni/eu@nfOP under AM 1.5G illumination (100 mW cm⁻²).

In accordance with the reviewer's comment, we performed the chronoamperometry measurement at the potential of shoulder peak (0.523 V vs. RHE) to see whether this peak is related with the redox potential of Ni-based OER catalyst or water oxidation. The photocurrent density gradually decreased and both O₂ and H₂ evolution were not detected. With these results above, we can conclude that the shoulder peak originated from redox current on Ni based OER catalyst as reported in the previous literatures (*J. Am. Chem. Soc.* 2014, 136, 6744–6753; *J. Am. Chem. Soc.* 2017, 139, 11361–11364) and was not related with water oxidation reaction (Review only Fig. 4)

Review only Fig. 4 | Chronoamperometry measurement at surface catalyst oxidation potential. a, Current density – potential curve of LDH/Ni/eu@nfOP. **b,** Current density – time curve of LDH/Ni/eu@nfOP at 0.523 V vs. RHE which is surface catalyst oxidation potential.

Third, in order to rule out the possibilities of side reactions such as photoanode oxidation or corrosion, we measured XPS, SEM, and UV-Vis spectra of the LDH/Ni passivated organic photoanode (LDH/Ni/eu@nfOP) before and after the photoelectrochemical water splitting reaction or light illumination. As responded to first reviewer's comment, there were no significant change on XPS, SEM, and UV-Vis spectra (under visible light illumination) indicating that there was no considerable oxidation or corrosion on photoanode during the photoelectrochemical water splitting.

Revised Supplementary Fig. 17 | High-resolution XPS spectrum of NiFe-LDH/Ni/eu@nfOP. a, b, Ni 2p, c, d, Fe 2p in LDH/Ni/eu@nfOP before and after 10 h stability test, respectively.

Revised Supplementary Fig. 18 | SEM images of LDH/Ni/eu@nfOP. a,b, Surface of LDH/Ni/eu@nfOP before and c,d, after 10 h stability test.

Revised Supplementary Fig. 19 | Comparison of UV-Vis absorption spectra of PBDB-T:ITIC-based BHJ layer before and after 10 h light irradiation. a, Under irradiation of AM 1.5G illumination (UV + Vis) light. **b,** Under irradiation of only Vis light.

Manuscript text, Page no. 15

“However, when there exists sulfite in the solution, the stability of Ni/eu@nfOP is significantly improved with the increased surface charge separation efficiency (Supplementary Fig. 15 and 16. To rule out the possibilities of other reactions such as surface oxidation of LDH/Ni/eu@nfOP during water splitting reactions, we performed X-ray photoelectron spectroscopy (XPS) and scanning electron microscopy (SEM) before and after the photoelectrochemical water splitting reaction. (Supplementary Fig. 17). The high resolution Ni 2p and Fe 2p spectra regions show splitting into 2p_{3/2} and 2p_{1/2} doublets due to spin-orbit coupling⁴⁵. The binding energy peaks of Ni 2p_{3/2} and 2p_{1/2} are located at 855.1 eV and 872.6 eV (Supplementary Fig. 17a, b), respectively, which fit Ni²⁺^{47,48,53,54}. Furthermore, the XPS peaks at binding energies of 712.3 eV and 724.1 eV correspond to Fe 2p_{3/2} and 2p_{1/2}, indicating the +3 oxidation state of Fe^{47,53,54} (Supplementary Fig. 17c, d). When comparing of XPS spectra of LDH/Ni/eu@nfOP before and after the 10 h stability test, no appreciable change in binding energy is shown. Also, no difference in surface morphology of LDH/Ni/eu@nfOP between before and after the 10 h stability test is verified by SEM observations (Supplementary Fig. 18).”

Manuscript text, Page no. 16

“We can also observe that there was negligible decrease, and 15% of decrease in the absorbance of organic photoactive layers after 10 hours of Vis light irradiation (420 nm cut-off filter, 100 mW cm⁻²) and AM 1.5G illumination (UV+Vis, 100 mW cm⁻²), respectively (Supplementary Fig. 19).”

Comment 2: While the use of the eutectic is not a new concept, the PBDB-T:ITIC bulk heterojunction is novel in the field of polymer photoelectrochemical cells (anodes or cathodes). It is striking though, that the authors entirely omit citations of previous works on polymer photoelectrochemical cells. Admittedly, only few works exist demonstrating polymer photoanodes (i.e. BBL: Bornoz et al., *J Am Chem Soc.* 2015, 137, 15338–15341), but there are many works on polymer photocathodes including the works on organic photovoltaics used in electrolyte solutions (i.e. Bourgeteau et al., *ACS Appl. Mater. Interfaces* 2015, 7, 16395-16403) and others cited in recent reviews (Steier, Holiday, *J. Mater. Chem. A*, 2018, 6, 21809-21826; Bellani, Antognazza, Bonaccorso, *Adv. Mater.* 2019, 31, 1801446.) I encourage the authors to cite previous works on polymer materials and bulk heterojunctions used in photoelectrochemical water splitting to put their work into context.

Response: We thank the reviewer for the critical and significant comments. In accordance with the reviewer's recommendation, we referred the context related with the previous literature dealing with other organic-based photoelectrochemical cells.

Added references

- 16 Steier, L. & Holliday, S. *A bright outlook on organic photoelectrochemical cells for water splitting. J. Mater. Chem. A* 6, 21809-21826 (2018).
- 17 Bellani, S., Antognazza, M. R. & Bonaccorso, F. *Carbon-based photocathode materials for solar hydrogen production. Adv. Mater.* 31, 1801446 (2019).
- 18 Otep, S., Michinobu, T. & Zhang, Q. *Pure organic semiconductor-based photoelectrodes for water splitting. Solar RRL*, 1900395 (2019).
- 19 Bourgeteau, T. et al. *A H₂-evolving photocathode based on direct sensitization of MoS₃ with an organic photovoltaic cell. Energy Environ. Sci.* 6, 2706-2713 (2013).
- 20 Bourgeteau, T. et al. *Enhancing the performances of P3HT:PCBM–MoS₃-based H₂-evolving photocathodes with interfacial layers. ACS Appl. Mater. Interfaces* 7, 16395-16403 (2015).
- 21 Yao, L. et al. *Establishing stability in organic semiconductor photocathodes for solar hydrogen production. J. Am. Chem. Soc.* 142, 7795-7802 (2020).
- 22 Shirakawa, H., Ikeda, S., Aizawa, M., Yoshitake, J. & Suzuki, S. *Polyacetylene film: A new electrode material for photoenergy conversion. Synthetic Metals* 4, 43-49 (1981).
- 23 Bornoz, P., Prévot, M. S., Yu, X., Guijarro, N. & Sivula, K. *Direct light-driven water oxidation by a ladder-type conjugated polymer photoanode. J. Am. Chem. Soc.* 137, 15338-15341 (2015).
- 24 Dai, C., Gong, X., Zhu, X., Xue, C. & Liu, B. *Molecular modulation of fluorene-dibenzothiophene-S,S-dioxide-based conjugated polymers for enhanced photoelectrochemical water oxidation under visible light. Mater. Chem. Front.* 2, 2021-2025 (2018).
- 25 Ruan, Q. et al. *A nanojunction polymer photoelectrode for efficient charge transport and separation. Angew. Chem., Int. Ed.* 56, 8221-8225 (2017).
- 26 Peng, G., Volokh, M., Tzadikov, J., Sun, J. & Shalom, M. *Carbon nitride/reduced graphene oxide film with enhanced electron diffusion length: An efficient photoelectrochemical cell for hydrogen generation. Adv. Energy Mater.* 8, 1800566 (2018).

There are several organic photocathode works reported but only few organic photoanodes works have been reported, as the reviewer's comments (*J. Am. Chem. Soc.* 2015, 137, 15338 ; *Mater. Chem. Front.* 2018, 2, 2021; *Adv. Energy Mater.* 2018, 8, 1800566). However, in most cases of the organic photoanodes and photocathodes reported so far, they lost their performances in a few minutes and even for the recent stable organic photoelectrodes which were stabilized by TiO₂ layers, less than 50% initial photocurrent is maintained after 1 hour.

In contrast, in our work, 90% and 98% of the initial photocurrent densities were retained after 10 hours under solar simulated light and visible light, respectively, which shows the best stability among all the organic based photoelectrodes to our best knowledge. The reason for such high stability is that nickel foil perfectly protected the organic photoanode from water, and GaIn eutectic and NiFe layered double hydroxide catalysts helped efficient charge transfer for water oxidation. These strategies are also reasons for the higher performance of our organic photoelectrodes in comparison with those of the other group's works. The photocurrents at 1.23 V vs. RHE and half-STH values of other organic photoanodes were in the range of 7~100 $\mu\text{A cm}^{-2}$ and 0.0001~0.018% respectively which are about several orders lower than ours (15.1 mA cm^{-2} at 1.23 V vs. RHE and $\eta_{\text{half-STH}} = 4.65\%$). They directly applied organic activator on FTO, while we used the bulk heterojunction with hole and electron transfer layers. This difference also played an important role in such big performance difference between our study and literatures. We added other organic photoanode performance data in the revised Supplementary Figure 1 and revised Supplementary Table 1 for better comparisons.

Revised Supplementary Fig. 1 | Recent reported half-cell solar-to-hydrogen conversion efficiencies ($\eta_{\text{half-STH}}$) and photocurrent density at 1.23 V vs. reversible hydrogen evolution of photoanodes.

Revised Supplementary Table. 1 | Recent Rreported PEC performance of photoanodes (2017–2019).

Year	Photoanode	Onset potential (V vs. RHE)	J at 1.23 V vs. RHE (mA cm ⁻²)	$\eta_{half-STI}$ (%)	Electrolyte	Ref
2015	BBL	< 0.7	0.015	~0.03 %	0.5 M Na ₂ SO ₄ (pH 7)	(1)
2017	np ⁺ -Si/SiO _x /NiO _x /NiFe	0.89	30.7	3.30%	1M KOH (pH 13.7)	(2)
2017	NiOOH-FeOOH/CQDs/BiVO ₄	-	5.99	2.29%	potassium phosphate (pH 7)	(3)
2017	CoO _x coated Mo,N-BiVO ₄ on Ti substrate	-	5.04	1.41%	0.1M phosphate buffer (pH 7.4)	(4)
2017	CoPi/GaN/Ta ₃ N ₅	0.65	-	1.50%	0.5M potassium phosphate (pH 13)	(5)
2017	g-C ₃ N ₄	0.5	0.063	~0.014 %	0.1 M Na ₂ SO ₄ (pH 7)	(6)
2017	s-BCN	0.5	0.1023	~0.02 %	0.1 M Na ₂ SO ₄ (pH 6.6)	(6)
2018	NiFe-LDH onto the polycrystalline n ⁺ p-Si substrate	0.78	37	4.30%	1M KOH	(7)
2018	[Co ₂ (bim) ₄]-modified BiVO ₄	-	3.1	0.90%	0.5 M Na ₂ SO ₄ .	(8)
2018	β -FeOOH/BiVO ₄	-	4.3	0.71%	0.2 M Na ₂ SO ₄ (pH 7)	(9)
2018	Ti:Fe ₂ O ₃ @GCNN-CQDs	-	3.38	0.33%	1 M KOH (pH 13.6)	(10)
2018	CoOOH/(Ti, C) - Fe ₂ O ₃	-	1.85	0.11%	NaOH (pH 14)	(11)
2018	surface modified TiO ₂ -B/anatase core/shell NWs	-	1.69	1.36%	1M NaOH	(12)
2018	red-colored polymerized C ₃ N ₄ on TiO ₂ nanorod	-	2.33	0.63%	1M Na ₂ SO ₄	(13)
2018	rGO/g-C ₃ N ₄	0.8	0.072	~0.018 %	0.1 M KOH	(14)
2019	n-Si/SiO _x /Al ₂ O ₃ /Ni/NiO _x /NiOOH	0.85	28	3%	1M KOH	(15)
2019	black BiVO ₄ @TiO _{2-x}	-	6.12	2.50%	0.5M potassium phosphate (pH 7)	(16)
2019	Mo-BiVO ₄ @rGO composite	0.19	8.51	2.45%	0.1M Na ₂ SO ₄ (pH 7)	(17)
2019	Ti ³⁺ /Ni co-doped TiO ₂ nanotube	-	-	1.51%	1M KOH	(18)
2019	n-Si/CoO _x /NiCuO _x	1.04	16.6	1.42%	1M NaOH	(19)
2019	NiFeO _x /Ta ₃ N ₅ /GaN/Al ₂ O ₃	0.65	6.3	1.15%	0.2M potassium phosphate (pH 13)	(20)
2019	F/Mo:BiVO ₄ with CoPi	-	4.78(F) / 5.43(B)	1.10%	0.1M KH ₂ PO ₄ /K ₂ HPO ₄ buffer (pH 7.3)	(21)
2019	Ultrathin Co(OH) _x encap p-Cu ₂ S/n-BiVO ₄	-	3.51	0.94%	0.5M potassium phosphate (pH 10)	(22)
2019	NiO/BiVO ₄	-	2.75	0.72%	0.1M Potassium phosphate (pH 7.4)	(23)
2019	BiFeO ₃ coated Sn:TiO ₂ (BFO/Sn:TiO ₂)	0.18	1.47	0.72%	1M NaOH	(24)
2019	CoPi onto Mo:BiVO ₄	-	2.98	0.52%	0.1M phosphate buffer (pH 7)	(25)

2019	Fe ₂ O ₃ /TiO ₂	-	2.9	0.20%	1M KOH (26)
This work	Ni/eu@nfOP	0.63	14.7	3.53%	1M NaOH (pH 13.6)
This work	LDH/Ni/eu@nfOP	0.47	15.4 15.1	4.53% 4.65%	1M NaOH (pH 13.6)

Manuscript text, Page no. 4

~~“In addition, organic semiconductors can be used for both photoanodes and photocathodes by simply converting their order of deposition^{16,17}. Despite these merits, organic semiconductors have not attracted attention as photoactive materials for PEC water splitting due to their low stability in aqueous solutions¹⁸.”~~

→ “Several organic semiconductors were applied for photocathodes, since the first demonstration of photocathodic hydrogen production by Shirakawa et al. with polyacetylene in 1981¹⁹⁻²², and only few organic photoactive-layer-based photoanodes have been reported. For example, Bornož et al. and Dai et al. showed that the poly[benzimidazobenzophenanthroline] and fluorine-dibenzothiophene-S,S-dioxide-based conjugated polymer can be utilized as photoanodes for direct solar water oxidation, respectively^{23,24}. Ruan et al. and Peng et al. showed the possibility of usage of carbon nitride materials as photoanodes^{25,26}. However, these organic photoactive-layer-based photoanodes not only exhibited photocurrent densities of only few microampere scales at 1.23 V vs. RHE (~100 $\mu\text{A cm}^{-2}$) but half-cell solar-to-hydrogen conversion efficiencies ($\eta_{\text{half-STH}}$) were also lower than 0.03% so far (Supplementary Table 1 and Supplementary Fig. 1). Furthermore, they lost their performances in a few minutes and even for the case of the recent stable organic photoelectrodes stabilized by TiO₂ layers, only less than 50% initial photocurrent is maintained after 1 hour.¹⁸”

We greatly appreciate the first and second reviewers' critical comments on our manuscript. With reference to your advices, we were able to conduct a more complete study. Thank you for your consideration.

Ji-Wook Jang, PhD

Assistant Professor

School of Energy and Chemical Engineering

Ulsan National Institute of Science and Technology (UNIST)

Ulsan, Republic of Korea

Tel: +82 52 217 3027, E-mail: jiwjang@unist.ac.kr.

REVIEWER COMMENTS

Reviewer #1 (Remarks to the Author):

The reviewer has been surprised at the revised version. The revision was almost perfectly done and responded to the reviewers' comments. The reviewer appreciates the H₂/O₂ evaluations with high Faraday efficiency, which is not so easy experiments.

The structure of the working electrode is clearly shown in comparison to the previous version. (while the area of the electrode is not shown.)

1) The reviewer pointed out the high current efficiency 15.4mA/cm² is noteworthy, and the authors emphasize it in the revised version. However the current (also H₂/O₂ evolution) depends on the outer bias potential. The bias level of 1.3 V vs RHE means there are no contributions as up-hill reactions as shown in Figure 2f ($\eta \sim 0\%$) in the revised version.

I feel it would be much better to show the current, H₂, and O₂ amounts for the 0.8 V for LDH/Ni@NFOP and 0.9 V for Ni@NFOP because these potentials give the highest energy conversion efficiency. (Of course the shown data is useful to understand the scientific aspects.)

2) A more serious question is about the equation of η shown in page 13. The equation for the experiment is for a three-electrode system and the authors do not care about the potential of the counter electrode. If so, η is only for oxygen evolution, while the H₂ from the counter electrode was uncontrolled by the bias potential. If the authors think of practical application, the two-electrode system should be considered especially for the discussion about the efficiency.

3) As for the citation to add organic photoelectrochemistry in the revised version, the reviewer feels 3 points.

3-1) Inorganic PEC is not surveyed. Nowadays, the highest efficiency has been shown for non-oxide inorganic systems for example Prof. Domen's Group.

3-2) page 4, ref16-18, "the simply converting their order of deposition" was shown in Prof. Abe's paper as below.

J. Electroanal. Chem, 583 (2), 327-332, (2005).

3-3) page 4 line 5, As for the "organic active layer based photoanode", the following paper is much more leading than the cited examples.

Angew. Chem., 118 (17), 2844-2847, (2006).

4) small questions,

4-1) page 19 line 1. The intensity 100 mW/cm² is after the filter or before?

4-2) Figure 3a, the time dependent (days) E-V curves means the change due to only immersion of the electrode? Does the days mean the illumination time?

4-3) Figure 3c, I am curious about the absolute values of the currents for UV+vis vs vis.

4-4) Page 20, line 9, show the area of the electrode.

Reviewer #2 (Remarks to the Author):

I am pleased with the additional experiments confirming quantitative gas evolution at the photoanode and the more detailed analysis of the onset potential verifying oxidation of Ni species which does not contribute to O₂ evolution.

In the revised manuscript key literature on organic photoelectrochemical cells is now acknowledged and discussed. In the context of the body of existing literature this work demonstrates a remarkable photoelectrochemical water oxidation performance with >10 h stable polymer photoanodes and provides key experimental evidence for their claims.

I have only one last request of clarifying the experimental section on the electrode fabrication. With

the additional photographs provided (Supplementary Fig. 3) it is not clear when the epoxy is actually applied because in the methods it seems epoxy is applied after the NiFeLDH-Ni foil is coated on the eutectic metal whereas in Suppl. Fig. 3 it seems it is applied before the eutectic is coated. Please clarify this part. Please also specify the thickness of the Ni foil.

Response to Comments of Reviewer #1

General Comment: The reviewer has been surprised at the revised version. The revision was almost perfectly done and responded to the reviewers' comments. The reviewer appreciates the H₂/O₂ evaluations with high Faraday efficiency, which is not so easy experiments.

The structure of the working electrode is clearly shown in comparison to the previous version. (while the area of the electrode is not shown.)

Response: We thank the reviewer for the comments. We appreciate your kindness and very positive opinions about our experimental results. We would like to carefully address your comments one by one.

Comment 1: The reviewer pointed out the high current efficiency 15.4mA/cm² is noteworthy, and the authors emphasize it in the revised version. However the current (also H₂/O₂ evolution) depends on the outer bias potential. The bias level of 1.3 V vs RHE means they are no contribution as up-hill reaction as shown in the Figure 2f ($\eta \sim 0\%$) in the revised version.

I feel it would be much better to show the current, H₂, and O₂ amounts for the 0.8 V for LDH/Ni/eu@nfOP and 0.9 V for Ni/eu@nfOP because this potential gives the highest energy conversion efficiency. (Of course the shown data is useful to understand the scientific aspects.)

Response: We thank the reviewer for the valuable comment. We totally agree with the reviewer's comment that we need to measure the performance at the potentials where they shows the best conversion efficiencies. We measured the photocurrents and the Faradaic efficiencies of H₂ and O₂ evolution at 0.8 V and 0.9 V vs. RHE for LDH/Ni/eu@nfOP and Ni/eu@nfOP, respectively. Their Faradaic efficiencies of H₂ and O₂ production were almost 100% with the H₂/O₂ ratio of 2 (revised Supplementary Information 10, Reviewer only Fig. 1)

Supplementary Fig. 10 | The photoelectrochemical water splitting at highest $\eta_{half-STH}$ potential. a, The current density – time curves of LDH/Ni/eu@nfOP at 0.8 V vs. RHE for water oxidation in 1 M NaOH under AM 1.5 G illumination. **b,** Faradaic efficiencies for H₂ and O₂ production and H₂/O₂ ratio for LDH/Ni/eu@nfOP at the same condition.

Reviewer only Fig. 1 | The photoelectrochemical water splitting at highest $\eta_{half-STH}$ potential. a, Current density – time curves of Ni/eu@nfOP at 0.9 V vs. RHE for water oxidation in 1 M NaOH under AM 1.5 G illumination. **b,** Faradaic efficiencies for H₂ and O₂ production and H₂/O₂ ratio for Ni/eu@nfOP at the same condition.

Manuscript text, Page no. 15

“The Faradaic efficiencies for PEC H₂ and O₂ evolution of LDH/Ni/eu@nfOP are near 100% and the H₂/O₂ molar ratio is also closed to 2 (Supplementary Fig. 9 and Fig. 2h).”

→ “The Faradaic efficiencies for PEC H₂ and O₂ evolution of LDH/Ni/eu@nfOP at 0.8 V and 1.3 V vs. RHE are near 100% and their H₂/O₂ molar ratio are also closed to 2 (Fig. 2h, Supplementary Fig. 9 and 10).”

Comment 2: A more serious question is about the equation of η shown in the page 13. The equation the experiment is for the three electrode system and the authors does not care about the potential of the counter electrode. If so, η is only for oxygen evolution, while the H₂ from the counter electrode was uncontrolled bias potential. If the authors think the practical application, the two electrode system should be considered especially for the discussion about the efficiency.

Response: We thank the reviewer for the critical comment. We totally agreed with the reviewer’s comments that the measurement of half-cell solar-to-hydrogen conversion efficiency ($\eta_{half-STH}$) in a two-electrode configuration is more accurate than that measured in a three-electrode configuration. We changed all the $\eta_{half-STH}$ values obtained in the three-electrode configuration (4.63%) to $\eta_{half-STH}$ obtained in a two-electrode configuration (4.33%) in the revised manuscript including abstract, introduction, result, conclusion part and related Figures and Table (Revised Fig. 2f, Supplementary Fig. 7, Supplementary Fig. 1 and Supplementary Table. 1). We didn’t included the $\eta_{half-STH}$ of Ni/eu@nfOP, because it is not a best record value.

Revised Figure 2f, $\eta_{\text{half-STH}}$ of LDH/Ni/eu@nfOP measured in a two-electrode configuration.

Supplementary Fig. 7 | Current density–potential of LDH/Ni/eu@nfOP in a two-electrode system.

Manuscript text, Page no. 14

~~“where V_{app} and J_{op} are the applied external bias voltage vs. RHE in a three electrode configuration and photocurrent density of a photoelectrode under 1 sun illumination, respectively.”~~

→ “where V_{app} and J_{op} are the applied external bias voltage vs. counter electrode (Pt) in a two-electrode configuration and photocurrent density of a photoelectrode under 1 sun illumination, respectively.”

Manuscript text, Page no. 14

~~“The $\eta_{\text{half-STH}}$ values of Ni/eu@nfOP and LDH/Ni/eu@nfOP are 3.53% and 4.65% at 0.79 and 0.87 V vs. RHE, respectively (Fig. 2f). The $\eta_{\text{half-STH}}$ of 4.65% is the highest value among those of all reported photoanodes, including metal oxide and silicon based materials, to the best of our knowledge (Supplementary Table 1).”~~

→ “The $\eta_{\text{half-STH}}$ values of LDH/Ni/eu@nfOP is 4.33% at 0.82 V vs. Pt (Fig. 2f and Supplementary Fig. 7), which is the highest value among those of all reported photoanodes, including metal-oxide-, metal-nitride-, silicon-, and other organic semiconductor-based materials, to the best of our knowledge (Supplementary Table 1).”

Manuscript text, Page no. 14

~~“The $\eta_{\text{half-STH}}$ obtained from a two-electrode configuration is 4.33%, which is similar to that obtained by the three-electrode configuration (Supplementary Fig. 7).”~~

Comment 3: As for the citation to add organic photo electrochemistry in the revised version, the reviewer feels 3 points.

- 1) Inorganic PEC is not surveyed. Nowadays, highest efficiency has been shown for the non-oxide inorganic systems for example Prof. Domen's Group.
- 2) page 4, ref16-18, "the simply converting their order of deposition" was shown in Prof. Abe's paper as below.
J. Electroanal. Chem, 583 (2), 327-332, (2005).
- 3) page 4 line 5, As for the "organic active layer based photoanode", the following paper is much more leading paper than the cited examples.
Angew. Chem., 118 (17), 2844-2847, (2006).

Response: We thank the reviewer for the valuable comments. In accordance with the reviewer's recommendation, we cited the above literatures and added relevant explanation in the revised manuscript.

Comment 1)

Inorganic PEC is not surveyed. Nowadays, highest efficiency has been shown for the non-oxide inorganic systems for example Prof. Domen's Group.

Response 1)

We thank the reviewer for the important comments about addition of non-oxide inorganic PEC as representative examples of inorganic semiconductors. In accordance with the reviewer's comment, we added several examples of non-oxide inorganic systems (tantalum nitride based photoelectrode) to the revised Supplementary Fig. 1 and revised Supplementary Table 1 with the relevant explanation.

Revised supplementary Fig. 1 | Recent reported half-cell solar-to-hydrogen conversion efficiencies ($\eta_{\text{half-STH}}$) and photocurrent density at 1.23 V vs. reversible hydrogen evolution of photoanodes.

Supplementary Table. 1 | Recent reported PEC performance of photoanodes.

Year	Photoanode	Onset potential (V vs. RHE)	J at 1.23 V vs. RHE (mA cm ⁻²)	$\eta_{half-STI}$ (%)	Electrolyte	Ref
2013	Co-Pi/Ba-Ta ₃ N ₅	0.65	6.7	1.5	0.5 M K ₂ HPO ₄ (pH 13)	(1)
2015	BBL	< 0.7	0.015	~0.03 %	0.5 M Na ₂ SO ₄ (pH 7)	(2)
2016	Integrated Ta ₃ N ₅ (P)	0.6	12.1	2.5	1 M NaOH (pH 13)	(3)
2017	np ⁺ -Si/SiO _x /NiO _x /NiFe	0.89	30.7	3.30%	1M KOH (pH 13.7)	(4)
2017	NiOOH-FeOOH/CQDs/BiVO ₄	-	5.99	2.29%	potassium phosphate (pH 7)	(5)
2017	CoO _x coated Mo,N-BiVO ₄ on Ti substrate	-	5.04	1.41%	0.1M phosphate buffer (pH 7.4)	(6)
2017	CoPi/GaN/Ta ₃ N ₅	0.65	-	1.50%	0.5M potassium phosphate (pH 13)	(7)
2017	g-C ₃ N ₄	0.5	0.063	~0.014 %	0.1 M Na ₂ SO ₄ (pH 7)	(8)
2017	s-BCN	0.5	0.1023	~0.02 %	0.1 M Na ₂ SO ₄ (pH 6.6)	(8)
2018	NiFe-LDH onto the polycrystalline n ⁺ p-Si substrate	0.78	37	4.30%	1M KOH	(9)
2018	[Co ₂ (bim) ₄]-modified BiVO ₄	-	3.1	0.90%	0.5 M Na ₂ SO ₄ .	(10)
2018	β -FeOOH/BiVO ₄	-	4.3	0.71%	0.2 M Na ₂ SO ₄ (pH 7)	(11)
2018	Ti:Fe ₂ O ₃ @GCNN-CQDs	-	3.38	0.33%	1 M KOH (pH 13.6)	(12)
2018	CoOOH/(Ti, C) - Fe ₂ O ₃	-	1.85	0.11%	NaOH (pH 14)	(13)
2018	surface modified TiO ₂ -B/anatase core/shell NWs	-	1.69	1.36%	1M NaOH	(14)
2018	red-colored polymerized C ₃ N ₄ on TiO ₂ nanorod	-	2.33	0.63%	1M Na ₂ SO ₄	(15)
2018	rGO/g-C ₃ N ₄	0.8	0.072	~0.018 %	0.1 M KOH	(16)
2019	n-Si/SiO _x /Al ₂ O ₃ /Ni/NiO _x /NiOOH	0.85	28	3%	1M KOH	(17)
2019	black BiVO ₄ @TiO _{2-x}	-	6.12	2.50%	0.5M potassium phosphate (pH 7)	(18)
2019	Mo-BiVO ₄ @rGO composite	0.19	8.51	2.45%	0.1M Na ₂ SO ₄ (pH 7)	(19)
2019	Ti ³⁺ /Ni co-doped TiO ₂ nanotube	-	-	1.51%	1M KOH	(20)
2019	n-Si/CoO _x /NiCuO _x	1.04	16.6	1.42%	1M NaOH	(21)
2019	NiFeO _x /Ta ₃ N ₅ /GaN/Al ₂ O ₃	0.65	6.3	1.15%	0.2M potassium phosphate (pH 13)	(22)
2019	F/Mo:BiVO ₄ with CoPi	-	4.78(F) / 5.43(B)	1.10%	0.1M KH ₂ PO ₄ /K ₂ HPO ₄ buffer (pH 7.3)	(23)
2019	Ultrathin Co(OH) _x encap p-Cu ₂ S/n-BiVO ₄	-	3.51	0.94%	0.5M potassium phosphate (pH 10)	(24)
2019	NiO/BiVO ₄	-	2.75	0.72%	0.1M Potassium phosphate (pH 7.4)	(25)

2019	BiFeO ₃ coated Sn:TiO ₂ (BFO/Sn:TiO ₂)	0.18	1.47	0.72%	1M NaOH	(26)
2019	CoPi onto Mo:BiVO ₄	-	2.98	0.52%	0.1M phosphate buffer (pH 7)	(27)
2019	Fe ₂ O ₃ /TiO ₂	-	2.9	0.20%	1M KOH	(28)
2020	Ta ₃ N ₅ -NRs/BaTaO ₂ N/FeNiO _x	>0.6	4.7	0.45	0.5 M K ₂ HPO ₄ (pH 13)	(29)
2020	Ta ₃ N ₅ -NRs/FeNiO _x	0.6	3.1	0.58	0.5 M K ₂ HPO ₄ (pH 13)	(29)
This work	Ni/eu@nfOP	0.63	14.7	3.53%	1M NaOH (pH 13.6)	
This work	LDH/Ni/eu@nfOP	0.47 0.55	15.1	4.65% 4.33%	1M NaOH (pH 13.6)	

Manuscript text, Page no. 3

~~“Since the first successful demonstration of water splitting with a TiO₂ photoanode by Honda and Fujishima in 1972³, extensive research has been focused on strategies to maximize the STH efficiency of inorganic photocatalysts, such as SrTiO₃, NaTaO₃, WO₃, Fe₂O₃, and BiVO₄, because they are typically earth abundant compounds and are stable in water⁴⁻⁷.”~~

→ “Since the first successful demonstration of water splitting with a TiO₂ photoanode by Honda and Fujishima in 1972³, extensive research has been focused on strategies to maximize the STH efficiency of inorganic photocatalysts, such as SrTiO₃, NaTaO₃, WO₃, Fe₂O₃, BiVO₄, and Ta₃N₅, because they are typically earth-abundant compounds and are stable in water⁴⁻¹⁰.”

Added references

- 8 Li, Y. et al. Cobalt phosphate-modified barium-doped tantalum nitride nanorod photoanode with 1.5% solar energy conversion efficiency. *Nat. Commun.* **4**, 2566 (2013).
- 9 Liu, G. et al. Enabling an integrated tantalum nitride photoanode to approach the theoretical photocurrent limit for solar water splitting. *Energy Environ. Sci.* **9**, 1327-1334 (2016).
- 10 Pihosh, Y. et al. Development of a core-shell heterojunction Ta₃N₅-nanorods/BaTaO₂N photoanode for solar water splitting. *ACS Energy Lett.* **5**, 2492-2497 (2020).

Comment 2)

page 4, ref16-18, "the simply converting their order of deposition" was shown in Prof. Abe's paper as below.

Response 2)

In accordance with the reviewer's comments, we cited the following reference in the revised manuscript.

Added reference

- 22 Abe, T., Nagai, K., Sekimoto, K., Tajiri, A. & Norimatsu, T. Novel photocathodic characteristics of organic bilayer composed of a phthalocyanine and a perylene derivative in a water phase containing a redox molecule. *Journal of Electroanalytical*

Comment 3)

page 4 line 5, As for the "organic active layer based photoanode", the following paper is much more leading paper than the cited examples.

Angew. Chem., 118 (17), 2844-2847, (2006).

Response 3)

We thank the reviewer for suggesting the important paper which needs to be cited. We referred the paper with the relevant explanation in the revised manuscript.

Manuscript text, Page no. 4

"For example, Abe et al. showed the possibility that organic semiconductor can be applied for a photoanode with a composite of 3,4,9,10-perylenetetracarboxylic acid bisbenzimidazole (n-type semiconductor) and cobalt(II) phthalocyanine(p-type semiconductor)²⁷."

Added reference

27 Abe, T. et al. An organic photoelectrode working in the water phase: visible-light-induced dioxygen evolution by a perylene derivative/cobalt phthalocyanine bilayer. *Angew. Chem., Int. Ed.* 45, 2778-2781 (2006).

Comment 4: small questions,

- 1) page 19 line 1. The intensity 100 mW/cm² is after the filter or before?
- 2) Figure 3a, the time dependent (days) E-V curves means the change due to only immersion of the electrode? Does the days mean the illumination time?
- 3) Figure 3c, I am curious the absolute values of the currents for UV+vis vs vis
- 4) Page. 20, line 9, show the area of the electrode.

Response: We thank the reviewer for the reviewer's detailed comments on our paper. We would like to reply to the reviewer's comments one by one.

Comment 1)

page 19 line 1. The intensity 100 mW/cm² is after the filter or before?

Response 1)

During the experiment, we measured light intensity (100 mW/cm²) after it passed through the filter and lens. We wrote the relevant comment in the Method part.

Manuscript text, Page no. 21 (Methods)

"The size of the light absorber of the organic photoanodes was 0.5 cm², and the light was illuminated with a 300 W Xe arc lamp (Newport, 66902) with an air mass 1.5 global (AM 1.5G) filter, collimating lens, and infrared filter (water)."

Comment 2)

Figure 3a, the time dependent (days) E-V curves means the change due to only immersion of the electrode? Does the days mean the illumination time?

Response 2)

We thank the reviewer for the important question. We measured the E-V curves just after immersion of the electrodes in NaOH electrolyte without illumination or applying bias to check the chemical stability of our organic photoanode in alkali electrolyte. In order to make the explanation more clear, we revised the manuscript as follows.

Manuscript text, Page no. 15

*“First, to reveal the effect of chemical stability of LDH/Ni/eu@nfOP on the PEC performance, we measure J–E performances for five consecutive days **after the immersion of the electrode in 1M NaOH electrolyte for a day without light illumination.**”*

Comment 3)

Figure 3c, I am curious the absolute values of the currents for UV+vis vs vis

Response 3)

The absolute current values of LDH/Ni/eu@nfOP for UV+vis (Fig. 3c) are shown in Fig 2g. And the absolute value of currents for visible light is presented in Supplementary Fig 21. In order to remove the ambiguity, we revised the related contents as follows.

Manuscript text, Page no. 17

~~*“To assess photostabilities of our passivated organic materials, we perform a stability test under AM 1.5G illumination or Vis light (Fig. 3c and Supplementary Fig. 20).”*~~

→ “To assess photostabilities of our passivated organic materials, we perform a stability test under Vis light (Fig. 3c and Supplementary Fig. 21).”

Comment 4)

Page. 20, line 9, show the area of the electrode.

Response 4)

We thank the reviewer for the very important comment. The area of all the organic photoanodes in this work is 0.5 cm². We added photoanode area information in the revised manuscript.

Manuscript text, Page no. 8

~~*“The photocurrent density is 14.7 mA cm⁻² at 1.23 V vs. RHE with an onset potential of 0.63 V vs. RHE.”*~~

→ “The photocurrent density is 14.7 mA cm⁻² at 1.23 V vs. RHE with an onset potential of 0.63 V vs. RHE (the area of all the photoelectrodes is 0.5 cm²).”

Manuscript text, Page no. 20 (Methods)

~~“The patterned ITO glass substrates ($15 \Omega^{-1}$) were cleaned by ultrasonic treatment in detergent, distilled water, acetone, and isopropyl alcohol, and then dried in an oven overnight at 70 °C.”~~
→ “The patterned ITO glass substrates ($15 \Omega^{-1}$, $1.5 \times 1.5 \text{ cm}^2$) were cleaned by ultrasonic treatment in detergent, distilled water, acetone, and isopropyl alcohol, and then dried in an oven overnight at 70 °C.”

Manuscript text, Page no. 20 (Methods)

~~“Finally, 10 nm thick MoO_3 and 100 nm thick gold films were thermally evaporated under vacuum ($<0.5 \times 10^{-5} \text{ Pa}$).”~~
→ “Finally, 10-nm-thick MoO_3 and 100-nm-thick gold films (active area : 0.5 cm^2) were thermally evaporated under vacuum ($<0.5 \times 10^{-5} \text{ Pa}$).”

Manuscript text, Page no. 21 (Methods)

~~“Finally, an epoxy bond was applied to fix and encapsulate the electrode and dried at room temperature overnight (active area: 0.5 cm^2).”~~

Response to Comments of Reviewer #2

General Comment: I am pleased with the additional experiments confirming quantitative gas evolution at the photoanode and the more detailed analysis of the onset potential verifying oxidation of Ni species which does not contribute to O₂ evolution.

In the revised manuscript key literature on organic photoelectrochemical cells is now acknowledged and discussed. In the context of the body of existing literature this work demonstrates a remarkable photoelectrochemical water oxidation performance with >10 h stable polymer photoanodes and provides key experimental evidence for their claims.

Response: We thank the reviewer for the comments. We appreciate your kindness and very positive opinions about our experimental results. We would like to carefully address the reviewer's comments.

Comment 1: I have only one last request for clarifying the experimental section on the electrode fabrication. With the additional photographs provided (Supplementary Fig. 3) it is not clear when the epoxy is actually applied because in the methods it seems epoxy is applied after the NiFe LDH-Ni foil is coated on the eutectic metal whereas in Suppl. Fig. 3 it seems it is applied before the eutectic is coated. Please clarify this part. Please also specify the thickness of the Ni foil.

Response: We thank the reviewer for the valuable comment. We agree with the reviewer's comment that the organic photoanode fabrication process is not clearly explained in the manuscript. We applied epoxy bond two times. First, we use epoxy and silver paste to connect the copper wire. Second, after GaIn eutectic metal loading followed by LDH deposited Ni foil (NiFe-LDH/Ni foil) loading, we used epoxy again to fix NiFe-LDH/Ni foil and to encapsulate the electrode. We modified the fabrication scheme and added relevant explanation in the Figure caption of revised Supplementary Fig. 3 and Methods part. We also added information of the thickness of Ni foil (100 μm) in the revised manuscript.

Revised supplementary Fig. 3 | Schematic and photographic images of organic photoanode fabrication process. First, silver paste and epoxy were used to connect a copper wire on the prepared organic-photoactive-layer-based cell. Second, after applying GaIn eutectic, NiFe-LDH/Ni foil was loaded on the electrode. Finally, an epoxy bond was applied to fix NiFe-LDH/Ni foil and encapsulate the electrode.

Manuscript text, Page no. 6

~~“The detailed process for the photoelectrode fabrication is schematically illustrated in the relevant photographs of Supplementary Fig. 3.”~~

→ “The detailed process for the photoelectrode fabrication is schematically illustrated with the relevant photographs and explanations in Supplementary Fig. 3 and the Methods part.”

Manuscript text, Page no. 20 (Methods)

“After connecting a copper wire to the prepared OPVs with silver paste and epoxy, a NiFe-LDH/Ni foil was loaded on the electrode with GaIn eutectic (Aldrich, 99.99%) between them. Finally, an epoxy bond was applied to fix and encapsulate the electrode and dried at room temperature overnight (active area: 0.5 cm²).”

Manuscript text, Page no. 20 (Methods)

~~“Nickel foil (Alfa Aesar, 99.5%) was cleaned by ultrasonication in acetone, isopropyl alcohol, and ethanol for 3 min each.”~~

→ “Nickel foil (Alfa Aesar, 99.5%, 100 μm thickness) was cleaned by ultrasonication in acetone, isopropyl alcohol, and ethanol for 3 min each.”

We greatly appreciate the first and second reviewers' critical comments on our manuscript. With reference to your advices, we were able to conduct a more complete study. Thank you for your consideration.

Ji-Wook Jang, PhD

Assistant Professor
School of Energy and Chemical Engineering
Ulsan National Institute of Science and Technology (UNIST)
Ulsan, Republic of Korea
Tel: +82 52 217 3027, E-mail: jiwjang@unist.ac.kr.

REVIEWERS' COMMENTS

Reviewer #1 (Remarks to the Author):

The manuscript has been revised properly. The reviewer recommends the paper to be published as the present form.

Response to comments of Reviewer #1

General Comment: The manuscript has been revised properly. The reviewer recommends the paper to be published as the present form.

Response: We appreciate the reviewer's positive comments. Thanks to the reviewers' previous valuable comments, our manuscript was much improved.